# Disentangling by Subspace Diffusion

**David Pfau, Irina Higgins, Aleksandar Botev, Sébastien Racanière**
DeepMind
London, UK
`{pfau, irinah, botev, sracaniere}@google.com`

## Abstract

We present a novel nonparametric algorithm for symmetry-based disentangling of data manifolds, the Geometric Manifold Component Estimator (GEOMANCER). GEOMANCER provides a partial answer to the question posed by Higgins et al. (2018): is it possible to learn how to factorize a Lie group solely from observations of the orbit of an object it acts on? We show that fully unsupervised factorization of a data manifold is possible *if* the true metric of the manifold is known and each factor manifold has nontrivial holonomy – for example, rotation in 3D. Our algorithm works by estimating the subspaces that are invariant under random walk diffusion, giving an approximation to the de Rham decomposition from differential geometry. We demonstrate the efficacy of GEOMANCER on several complex synthetic manifolds. Our work reduces the question of whether unsupervised disentangling is possible to the question of whether unsupervised metric learning is possible, providing a unifying insight into the geometric nature of representation learning.

## 1 Introduction

The ability to disentangle the different latent factors of variation in the world has been hypothesized as a critical ingredient in representation learning [1], and much recent research has sought a fully unsupervised way to learn disentangled representations from data [2–18]. Because the community has not settled on a definition of "disentangling", much of this work relies on heuristics and qualitative criteria to judge performance. For instance, datasets are often constructed by varying interpretable factors like object position, rotation and color and methods are judged by how well they recover these predefined factors [19, 20]. One commonly used definition is that a representation is disentangled if the data distribution can be modeled as a nonlinear transformation of a product of independent probability distributions [20]. This leads to a pessimistic result, that the different latent factors are not identifiable without side information or further assumptions.

To escape this pessimistic result, we can turn to a different, symmetry-based definition of disentangling [21], rooted in the Lie group model of visual perception [22–26]. Instead of a product of *distributions*, the symmetry-based approach to disentangling considers a representation disentangled if it matches the product of *groups* that define the symmetries in the world. If the actions that define the possible transformations in the world form a group $G = G_1 \times G_2 \times ... \times G_m$, then a representation is disentangled under this definition if it decomposes in such a way that the action of a single subgroup leaves all factors of the representation invariant except for one (See Supp. Mat., Sec. B for a formal definition).

The symmetry-based definition is appealing as it resolves an apparent contradiction in the parallelogram model of analogical reasoning [27]. For concepts represented as vectors $\mathbf{a}$, $\mathbf{b}$ and $\mathbf{c}$ in a flat space, the analogy $\mathbf{a} : \mathbf{b} :: \mathbf{c} : \mathbf{d}$ can be completed by $\mathbf{d} = \mathbf{b} + \mathbf{c} - \mathbf{a}$, as $\mathbf{d}$ completes the fourth corner of a parallelogram. This model has worked well in practice when applied to embeddings learned by deep neural networks for words [28, 29] and natural images [30, 31], and often matches human judgments [32]. But for many natural transformations, including 3D rotation, it is not possible to form a representation in a flat vector space such that vector addition corresponds to composition of transformations. Instead, obser-

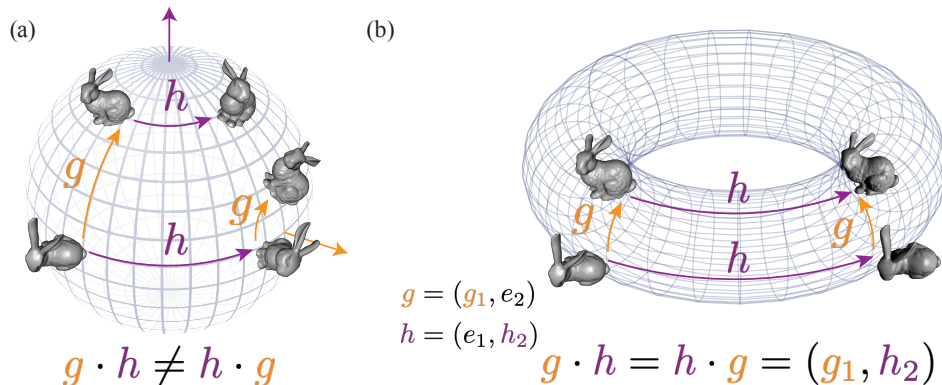

Figure 1: Two views of the orbit of an object under the action of a group, in this case images of the Stanford Bunny under changes in pose and illumination (a) Transformations from the same subgroup, in this case 3D rotations, do not in general commute, and the analogy is ambiguous. (b) Transformations from disentangled subgroups. The transformations commute and the analogy is unambiguous.

vations from such transformations can be naturally represented as coming from the orbit of a group [26]. In this setting, the parallelogram model breaks down – if $a,b,c$ are representations, and $g,h \in G$ are the group elements such that $g \cdot a = b$ and $h \cdot a = c$, then the completion of the analogy $a : b :: c : d$ is ambiguous, as $g \cdot h \cdot a \neq h \cdot g \cdot a$ for noncommutative operations (Figure 1). State-of-the-art generative models for single image classes typically elide this by restricting the dataset to a limited patch of the manifold, like forward-facing faces [33] or limited ranges of change in elevation [6, 34–38]. This ambiguity, however, is resolved if $G$ is a product of subgroups. So long as $g$ and $h$ leave all factors invariant except for one, and each varies a different factor, then $g$ and $h$ do commute, and the analogy can be uniquely completed.

While this definition of disentangling is appealing, it does not provide an obvious recipe for how to learn the appropriate factorization of the action on the world state. Some group invariances and equivariances can be built into neural network architectures [39–46], and it has been shown how commutative group representations can be learned [47]. Methods have been proposed to learn symmetry-based disentangled representations when conditioned on interactions [16, 17], or to learn dictionaries of Lie group operators from neighboring pairs of data [48, 49], but a general algorithm to factorize noncommutative groups without any supervision remains elusive.

If we restrict our attention to Lie groups – groups that are also manifolds, like rotations – we could use the properties of infinitesimal transformations as a learning signal. Essentially, we would like to use failures of the parallelogram model *as a learning signal itself*. Those directions that lie on disentangled submanifolds will behave like vectors in a flat space when one is moved in the direction of the other, hence complying with the parallelogram model, while directions within each submanifold may be mixed together in arbitrary ways. Computing all of the directions that remain invariant provides a disentangled factorization of the manifold. These intuitions can be made precise by the de Rham decomposition theorem [50], a foundational theorem in holonomy theory, a branch of differential geometry.

Here we present an algorithm that turns these ideas into a practical method for disentangling, the Geometric Manifold Component Estimator (GEOMANCER). GEOMANCER differs from other disentangling algorithms in that it does not learn a nonlinear embedding of the data. Instead, it can either be applied directly to the data so long as the local metric information is known, or it can be applied as a post-processing step to learned embeddings. GEOMANCER is a nonparametric algorithm which learns a set of subspaces to assign to each point in the dataset, where each subspace is the tangent space of one disentangled submanifold. This means that GEOMANCER can be used to disentangle manifolds for which there may not be a global axis-aligned coordinate system. GEOMANCER is also able to discover the correct number of dimensions in each submanifold without prior knowledge. Our algorithm is particularly well suited for dealing with transformations which have nontrivial holonomy, such as 3D rotations. In contrast, most previous work [2–16, 18] has focused on transformations with trivial holonomy, such as translation in 2D.

GEOMANCER builds on classic work on nonparametric manifold learning, especially Laplacian Eigenmaps [51], Diffusion Maps [52] and extensions like Orientable and Vector Diffusion Maps [53–55], generalizing the idea of finding modes of a random walk diffusion operator on manifolds

from scalars and vectors to matrices. These classic methods are primarily focused on learning a low-dimensional embedding of data. While GEOMANCER uses and extends many of these same mathematical techniques, its primary goal is to learn a factorization of the data manifold, which makes it an interesting new application of spectral graph theory.

GEOMANCER should not be confused with methods like Horizontal Diffusion Maps, which are for "synchronization" problems in manifold learning [56, 57]. These methods also use scalar and (co)vector diffusion, but not subspace diffusion. Much like other manifold learning problems, they are concerned with finding an embedding of points on a manifold, not a factorization. What distinguishes synchronization problems from the rest of manifold learning is that they are concerned with finding embeddings such that operations that relate pairs of data can be composed together in a cycle-consistent manner. GEOMANCER instead exploits the "cycle-inconsistency" of certain operations to distinguish entangled and disentangled directions around each point.

It is important to distinguish GEOMANCER from other manifold learning methods that go beyond learning a single undifferentiated manifold, especially Robust Multiple Manifold Structure Learning (RMMSL) [58]. RMMSL learns a *mixture* of manifolds which are all embedded in the same space. GEOMANCER by contrast, learns a single manifold which is itself a *product* of many submanifolds, where each submanifold exists in its own space. We next present a rapid overview of the relevant theory, followed by a detailed description of GEOMANCER, and finally show results on complex data manifolds.

## 2   Theory

We assume some basic familiarity with the fundamentals of Riemannian geometry and parallel transport, though we strongly encourage reading the review in Supp. Mat., Sec. A. For a more thorough treatment, we recommend the textbooks by Do Carmo [59] and Kobayashi and Nomizu [60]. Let $x$ denote points on the $k$-dimensional Riemannian manifold $\mathcal{M}$ with metric $\langle \cdot, \cdot \rangle_{\mathcal{M}}$, possibly embedded in $\mathbb{R}^n$. We denote paths by $\gamma : \mathbb{R} \to \mathcal{M}$, tangent spaces of velocity vectors by $T_x\mathcal{M}$, cotangent spaces of gradient operators by $T_x^*\mathcal{M}$, and vectors in $T_x\mathcal{M}$ by $\mathbf{v}$, $\mathbf{w}$, etc. As $\mathcal{M}$ is a Riemannian manifold, it has a unique Levi-Civita connection, which allows us to define a covariant derivative and parallel transport, which gives a formal definition of two vectors in nearby tangent spaces being parallel.

**Holonomy and the de Rham decomposition**
The basic tools of differential geometry, especially parallel transport, can be used to infer the product structure of the manifold through the *holonomy group*. Consider a loop $\gamma : [0,1] \to \mathcal{M}$, $\gamma(0) = x$, $\gamma(1) = x$. Given an orthonormal basis $\{\mathbf{e}_i\} \subset T_x\mathcal{M}$ of the tangent space at $x$, we can consider the parallel transport of all basis vectors around the loop $\{\mathbf{e}_i(t)\}$, illustrated in Fig. 2. Then the vectors $\{\mathbf{e}_i(1)\} \subset T_x\mathcal{M}$ form the columns of a matrix $H_\gamma$ that fully characterizes how any vector transforms when parallel transported around a loop. That is, for any parallel transport $\mathbf{v}(t)$ along $\gamma$, $\mathbf{v}(1) = H_\gamma \mathbf{v}(0)$. Note that if the affine connection preserves the metric, then $\{\mathbf{e}_i(t)\}$ will be an orthonormal basis for

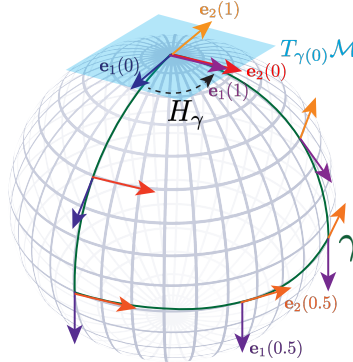

Figure 2: Holonomy on the sphere. The linear transform $H_\gamma$ captures the amount vectors in $T_{\gamma(0)}\mathcal{M}$ are rotated by parallel transport around the path $\gamma$.

all $t$, so $H_\gamma$ is in the orthonormal group $\mathrm{O}(k)$. Moreover if $\mathcal{M}$ is orientable, then the handedness of $\{\mathbf{e}_i(t)\}$ cannot change, so $H_\gamma$ is in the special orthonormal group $\mathrm{SO}(k)$, that is, $\det(H_\gamma) = 1$.

The linear transform $H_\gamma$ is the *holonomy* of the loop $\gamma$. The space of all holonomies for all loops that start and end at $x$ form the *holonomy group* $\mathrm{Hol}_x(\mathcal{M})$ at $x$. It can clearly be seen that this space is a group by considering different loops. The trivial loop $\gamma(t) = x$ has holonomy $I$, the identity. If $\gamma$ has holonomy $H_\gamma$, then the loop $\gamma'(t) = \gamma(1-t)$ has holonomy $H_\gamma^{-1}$, so if $H_\gamma$ is in the holonomy group, so is its inverse. And if $\gamma_1$ and $\gamma_2$ are loops, then the loop formed by first going around $\gamma_1$ followed by $\gamma_2$ has holonomy $H_{\gamma_2} H_{\gamma_1}$, so if two elements are in this group, then so is their product.

The structure of the holonomy group is extremely informative about the global structure of the manifold. If the manifold $\mathcal{M}$ is actually a product of submanifolds $\mathcal{M}_1 \times \mathcal{M}_2 \times \dots \times \mathcal{M}_n$, with the corresponding

product metric as its metric , then it is straightforward to show that  the tangent space $T_x\mathcal{M}$ can be decomposed into orthogonal subspaces $T_x^{(1)}\mathcal{M},...,T_x^{(m)}\mathcal{M}$ such that the action of $\mathrm{Hol}_x(\mathcal{M})$ leaves each subspace invariant. That is, if $\mathbf{v}\in T_x^{(i)}\mathcal{M}$, then $H_\gamma\mathbf{v}\in T_x^{(i)}\mathcal{M}$ for all $\gamma$ . These subspaces are each tangent to the respective submanifolds that make up $\mathcal{M}$. The more remarkable result is that the converse holds locally and, if the manifold is simply connected and geodesically complete, globally [50]. A manifold is simply connected if any closed loop can be continuously deformed into a single point, and it is geodesically complete if any geodesic can be followed indefinitely.

**Theorem 1.**  de Rham Decomposition Theorem (de Rham, 1952), see also [60, Theorem 6.1]: Assume $\mathcal{M}$ is a simply connected and geodesically complete Riemannian manifold.  If there exists a point $x\in\mathcal{M}$ and a proper subspace $U$ that is invariant under the action of the holonomy group $\mathrm{Hol}_x(\mathcal{M})$, then $\mathcal{M}$ is a product Riemannian manifold $\mathcal{M}_1\times\mathcal{M}_2$ with $T_x\mathcal{M}_1=U$ and $T_x\mathcal{M}_2=U^\perp$. The tangent spaces to $\mathcal{M}_1$ and $\mathcal{M}_2$ at any other point $y$ are obtained by parallel transporting $U$ and $U^\perp$ respectively along any path from $x$ to $y$.

The above theorem can be applied recursively, so that if the holonomy group leaves multiple pairwise orthogonal subspaces invariant, we can conclude that $\mathcal{M}$ is a product of multiple Riemannian manifolds. It seems quite remarkable that a property of the holonomy group in a single tangent space can tell us so much about the structure of the manifold. This is because the holonomy group itself integrates information over the entire manifold, so in a sense it is not really a local property at all.

This result is the main motivation for GEOMANCER – we aim to discover a decomposition of a data manifold by investigating its holonomy group.  The holonomy group is a property of *all possible paths*, so it cannot be computed directly. Instead, we build a partial differential equation that inherits properties of the holonomy group and work with a numerically tractable approximation to this PDE.

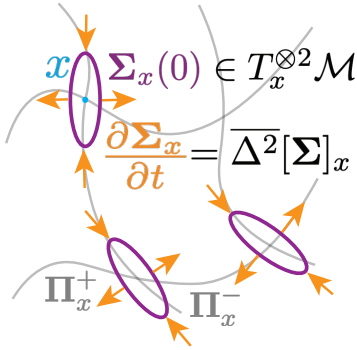

**Subspace diffusion on manifolds**    While it is not feasible to compute properties of all loops on a manifold, the *average* properties of random walk diffusion on a manifold can be computed by studying the properties of the diffusion equation. Consider a particle undergoing a Brownian random walk on a manifold with diffusion rate $\tau$. Then, given an initial probability density $p(x,0)$, the probability of finding the particle at $x$ at time $t$ evolves according to the diffusion equation:

$$\frac{\partial p(x,t)}{\partial t}=\tau\Delta^0[p](x,t) \tag{1}$$

Figure 3: Subspace diffusion. A field of symmetric semidefinite matrices $\mathbf{\Sigma}(t)$ evolves in time according to the differential equation $\dot{\mathbf{\Sigma}}=\overline{\Delta^2}[\mathbf{\Sigma}]$.

where $\Delta^0$ is a linear operator called the *Laplace-Beltrami* operator [60, Note 14, Vol. 2], defined as the trace of the second covariant derivative $\Delta^0[f]=\mathrm{Tr}\nabla^2 f$. Even if the initial condition is a delta function, the change in probability is nonzero everywhere, so the Laplace-Beltrami operator encodes global information about the manifold, though it weights local information more heavily.

The Laplace-Beltrami operator, which acts on scalar functions, can be generalized to the *connection Laplacian* for rank-$(p,q)$ tensor-valued functions. As the second covariant derivative of a rank-$(p,q)$ tensor is a rank-$(p,q+2)$ tensor, we can take the trace over the last two dimensions to get the connection Laplacian. The connection Laplacian also has an intuitive interpretation in terms of random walks of vectors. Given a probability density $p(\mathbf{v},t)$ over $T\mathcal{M}$, the manifold of all tangent spaces on $\mathcal{M}$ , the connection Laplacian on vectors $\Delta^1$ gives the rate of change of the mean of the density at every point $\mu_x=\int_{T_x\mathcal{M}}p(\mathbf{v},t)\mathbf{v}d\mathbf{v}$, while the connection Laplacian on matrices $\Delta^2$ gives the rate of change of the second moment $\mathbf{\Sigma}_x=\int_{T_x\mathcal{M}}p(\mathbf{v},t)\mathbf{v}\mathbf{v}^T d\mathbf{v}$ (Fig. 3).

Many of the properties of the holonomy group can be inferred from the second-order connection Laplacian $\Delta^2$. In particular, for a product manifold, the eigenfunctions of $\Delta^2$ contain information about the invariant subspaces of the holonomy group:

**Theorem 2.** Let $\mathcal{M}=\mathcal{M}_1\times...\times\mathcal{M}_m$ be a Riemannian product manifold, and let $T_x^{(1)}\mathcal{M},...,T_x^{(m)}\mathcal{M}$ denote orthogonal subspaces of $T_x\mathcal{M}$ that are tangent to each submanifold. Then the tensor fields

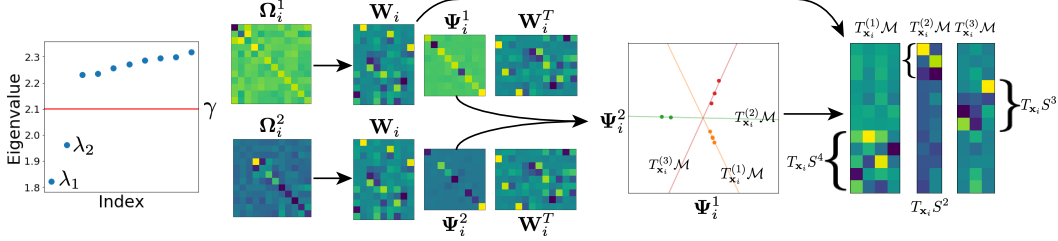

Figure 4: Illustration of the pipeline to go from eigenfunctions of $\overline{\Delta^2}$ to a set of bases for disentangled subspaces, shown on real data from the manifold $S^2 \times S^3 \times S^4$. Matrices $\boldsymbol{\Omega}_i^r$ up to the threshold in the spectrum $\gamma$ are simultaneously diagonalized as $\mathbf{W}_i \boldsymbol{\Psi}_i^r \mathbf{W}_i^T$. Columns of $\mathbf{W}_i$ are clustered based on the cosine similarity of the diagonals of $\boldsymbol{\Psi}_i^r$ to form bases for the subspaces $T_{\mathbf{x}_i}^{(1)} \mathcal{M}, \dots, T_{\mathbf{x}_i}^{(m)} \mathcal{M}$. For clarity, the data is visualized in the embedding space rather than coordinates of the tangent space.

$\boldsymbol{\Pi}^{(i)} : \mathcal{M} \to T_x \mathcal{M} \otimes T_x^* \mathcal{M}$ for $i \in 1, \dots, m$, where $\boldsymbol{\Pi}_x^{(i)}$ is the linear projection operator from $T_x \mathcal{M} \to T_x^{(i)} \mathcal{M}$, go to 0 under the action of the connection Laplacian $\Delta^2$.

We provide an informal argument and a more formal proof in Supp. Mat., Sec. B. There is an elegant parallel with the scalar Laplacian. The number of zero eigenvalues of the Laplacian is equal to the number of connected components of a graph or manifold, with each eigenvector being uniform on one component and zero everywhere else. For the second-order Laplacian, the zero eigenvalues correspond to factors of a product manifold, with the matrix-valued eigenfunction being the identity in the subspace tangent to one manifold and zero everywhere else. However, these are not in general the *only* eigenfunctions of $\Delta^2$ with zero eigenvalue, and we will discuss in the next section how GEOMANCER avoids spurious eigenfunctions.

## 3 Method

Here we describe the actual Geometric Manifold Component Estimator algorithm (GEOMANCER). The main idea is to approximate the second-order connection Laplacian $\Delta^2$ from finite samples of points on the manifold, and then find those eigenvectors with nearly zero eigenvalue that correspond to the disentangled submanifolds of the data. This allows us to define a set of *local* coordinates around every data point that are aligned with the disentangled manifolds.

Suppose we have a set of points $\mathbf{x}_1, \dots, \mathbf{x}_t \in \mathbb{R}^n$ sampled from some manifold embedded in $\mathbb{R}^n$. To construct $\Delta^2$, we first build up approximations to many properties of the manifold. In our discussion here, we will assume the data is embedded in a space where the $\ell_2$ metric in $\mathbb{R}^n$ matches the metric on the manifold. Such an embedding must exist for all manifolds [61], but how to learn it is an open question.

To start, we construct a symmetric nearest neighbors graph with edges $\mathcal{E} = \{e_{ij}\}$ between $\mathbf{x}_i$ and $\mathbf{x}_j$. This defines the set of possible steps that can be taken under a random walk. Next, we construct a set of tangent spaces, one per data point, by applying PCA to the difference between $\mathbf{x}_i$ and its neighbors $\mathbf{x}_j$ s.t. $e_{ij} \in \mathcal{E}$. The number of principal components $k$, equivalent to the dimensionality of the manifold, is a hyperparameter we are free to choose. This defines a set of local orthonormal coordinate systems $\mathbf{U}_i$ and local tangent vectors $\mathbf{v}_j$ s.t. $\mathbf{x}_j - \mathbf{x}_i \approx \mathbf{U}_i \mathbf{v}_j$ for neighboring points $\mathbf{x}_j$. This approach to constructing local tangent spaces is also used by many other manifold learning methods [53, 54, 62–64]. We will use these coordinates to construct the parallel transport from the point $\mathbf{x}_i$ to $\mathbf{x}_j$.

**Graph Connection Laplacians** To construct $\Delta^2$, we need a generalization of graph Laplacians to higher order tensors. The graph Laplacian is a linear operator on scalar functions on a graph, defined as:

$$\Delta^0[f]_i = \sum_{j \text{ s.t. } e_{ij} \in \mathcal{E}} f_i - f_j \tag{2}$$

Equivalently, if we represent functions over data points as a vector in $\mathbb{R}^t$ then the Laplacian can be given as a matrix $\Delta^0$ in $\mathbb{R}^{t \times t}$ with $\Delta_{ij}^0 = -1$ if $e_{ij} \in \mathcal{E}$ and $\Delta_{ii}^0 = n_i$ where $n_i$ is the number of neighbors of

---
**Algorithm 1:** Geometric Manifold Component Estimator (GEOMANCER)
---
**Data:** $\mathbf{x}_1,...,\mathbf{x}_t \in \mathbb{R}^n$ sampled from $\mathcal{M} = \mathcal{M}_1 \times ... \times \mathcal{M}_m$ with dimension $k$

**1. Build the manifold:**
    $e_{ij} \in \mathcal{E}$ if $\mathbf{x}_j \in \mathrm{knn}(\mathbf{x}_i)$ or $\mathbf{x}_i \in \mathrm{knn}(\mathbf{x}_j)$                  ▷ *Construct nearest neighbors graph*
    $d\mathbf{X}_i = (\mathbf{x}_{j_1} - \mathbf{x}_i, ..., \mathbf{x}_{j_{n_i}} - \mathbf{x}_i)$ for $j_1,...,j_{n_i}$ s.t. $e_{ij} \in \mathcal{E}$
    $\mathbf{U}_i \boldsymbol{\Sigma}_i \mathbf{V}_i^T = \mathrm{SVD}(d\mathbf{X}_i), T_{\mathbf{x}_i}\mathcal{M} \approx \mathrm{span}(\mathbf{U}_i)$         ▷ *Estimate tangent spaces by local PCA*

**2. Build and diagonalize the connection Laplacian:**
    $\mathbf{U}_{ij} \boldsymbol{\Sigma}_{ij} \mathbf{V}_{ij}^T = \mathrm{SVD}(\mathbf{U}_j^T \mathbf{U}_i)$
    $\mathbf{Q}_{ij} = \mathbf{U}_{ij} \mathbf{V}_{ij}^T$ for all $i,j$ s.t. $e_{ij} \in \mathcal{E}$               ▷ *Construct connection*
    $\underline{\Delta^2_{(ij)}} = -\mathbf{Q}_{ij} \otimes \mathbf{Q}_{ij}, \Delta^2_{(ii)} = n_i \mathbf{I}$     ▷ *Build blocks of 2nd-order graph connection Laplacian*
    $\overline{\Delta^2_{(ij)}} = \boldsymbol{\Pi}_{\mathrm{tr}}^T \boldsymbol{\Pi}_{\mathrm{sym}}^T \Delta^2_{(ij)} \boldsymbol{\Pi}_{\mathrm{sym}} \boldsymbol{\Pi}_{\mathrm{tr}}$     ▷ *Project blocks onto space of symmetric zero-trace matrices*
    $\overline{\Delta^2} \boldsymbol{\phi}^r = \lambda_r \boldsymbol{\phi}^r, r = 1,...,R$           ▷ *Compute bottom $R$ eigenfunctions/values of $\overline{\Delta^2}$*
    $\mathrm{vec}(\boldsymbol{\Omega}_i^r) = \boldsymbol{\Pi}_{\mathrm{sym}} \boldsymbol{\Pi}_{\mathrm{tr}} \boldsymbol{\phi}_i^r$             ▷ *Project eigenfunctions back to matrices*

**3. Align the results from different eigenvectors of the Laplacian:**
    $\mathbf{W}_i \boldsymbol{\Psi}_i^r \mathbf{W}_i^T = \boldsymbol{\Omega}_i^r$ for all $r$ s.t. $\lambda_r < \gamma$     ▷ *Simultaneously diagonalize matrices by* FFDIAG [67]
    $\boldsymbol{\psi}_{ik} = (\Psi_{i,kk}^1 ..., \Psi_{i,kk}^r, ..., \Psi_{i,kk}^{m-1})$
    $\mathcal{C}^j = \{\boldsymbol{\psi}_{ik} | \boldsymbol{\psi}_{ik}^T \boldsymbol{\psi}_{ik'} / ||\boldsymbol{\psi}_{ik}|| ||\boldsymbol{\psi}_{ik'}|| > 0.5\}$     ▷ *Cluster diagonals of $\boldsymbol{\Psi}_i$ by cosine similarity*
    $T_{\mathbf{x}_i}^{(j)}\mathcal{M} = \mathrm{span}(\{\mathbf{w}_{ik} | \boldsymbol{\psi}_{ik} \in \mathcal{C}^j\})$     ▷ *Columns of $\mathbf{W}_i$ in each cluster span the subspaces*

**Result:** Orthogonal subspaces $T_{\mathbf{x}_i}^{(1)}\mathcal{M}, ..., T_{\mathbf{x}_i}^{(m)}\mathcal{M}$ at every point $\mathbf{x}_i$ tangent to $\mathcal{M}_1, ..., \mathcal{M}_m$

---

$\mathbf{x}_i$. If the graph is approximating a Riemannian manifold, then in the limit of dense sampling the graph Laplacian becomes equivalent to the Laplace-Beltrami operator [65].

To generalize the graph Laplacian from scalars to vectors and tensors, we can replace the difference between neighboring scalars in Eqn. 2 with a difference between tensors. These must be tensors *in the same tangent space*, so any neighboring vectors must be parallel transported from $T_{\mathbf{x}_i}\mathcal{M}$ to $T_{\mathbf{x}_j}\mathcal{M}$, and similarly for higher-order tensors. On a graph, the parallel transport from $\mathbf{x}_i$ to $\mathbf{x}_j$ can be approximated by an orthonormal matrix $\mathbf{Q}_{ij}$ associated with $e_{ij}$, while the transport in the reverse direction is given by $\mathbf{Q}_{ji} = \mathbf{Q}_{ij}^T$. This leads to a natural definition for the first-order *graph connection Laplacian* [54]:

$$\Delta^1[\mathbf{v}]_i = \sum_{j \text{ s.t. } e_{ij} \in \mathcal{E}} \mathbf{v}_i - \mathbf{Q}_{ij}^T \mathbf{v}_j \tag{3}$$

This is a linear operator on vector-valued functions. We can represent vector-valued functions as a single flattened vector in $\mathbb{R}^{tk}$, in which case the graph connection Laplacian is a block-sparse matrix in $\mathbb{R}^{tk \times tk}$. Generalizing to matrices yields the second-order graph connection Laplacian:

$$\Delta^2[\boldsymbol{\Sigma}]_i = \sum_{j \text{ s.t. } e_{ij} \in \mathcal{E}} \boldsymbol{\Sigma}_i - \mathbf{Q}_{ij}^T \boldsymbol{\Sigma}_j \mathbf{Q}_{ij} \tag{4}$$

which, again flattening matrix-valued functions to vectors in $\mathbb{R}^{tk^2}$, gives a block-sparse matrix in $\mathbb{R}^{tk^2 \times tk^2}$. The diagonal blocks $\Delta^2_{(ii)} = n_i \mathbf{I}$ while the block for edge $e_{ij}$ is given by $\Delta^2_{(ij)} = -\mathbf{Q}_{ij} \otimes \mathbf{Q}_{ij}$.

While this gives a general definition of $\Delta^2$ for graphs, we still need to define the connection matrices $\mathbf{Q}_{ij}$. When a manifold inherits its metric from the embedding space, the connection is given by the projection of the connection in the embedding space. As the connection in Euclidean space is trivial, the connection on the manifold is given by the orthonormal matrix that most closely approximates the projection of $T_{\mathbf{x}_j}\mathcal{M}$ onto $T_{\mathbf{x}_i}\mathcal{M}$. In the local coordinates defined by $\mathbf{U}_i$ and $\mathbf{U}_j$, the projection is given by $\mathbf{U}_j^T \mathbf{U}_i$. If $\mathbf{U}_{ij} \boldsymbol{\Sigma}_{ij} \mathbf{V}_{ij}^T$ is the SVD of $\mathbf{U}_j^T \mathbf{U}_i$, then $\mathbf{Q}_{ij} = \mathbf{U}_{ij} \mathbf{V}_{ij}^T$ gives the orthonormal matrix nearest to $\mathbf{U}_j^T \mathbf{U}_i$. This is closely related to the canonical (or principal) angles between subspaces [66], and this connection was also used by Singer and Wu [54] for the original graph connection Laplacian.

**Eliminating Spurious Eigenfunctions**    We now have all the ingredients needed to construct $\Delta^2$. However, a few modifications are necessary to separate out the eigenfunctions that are projections

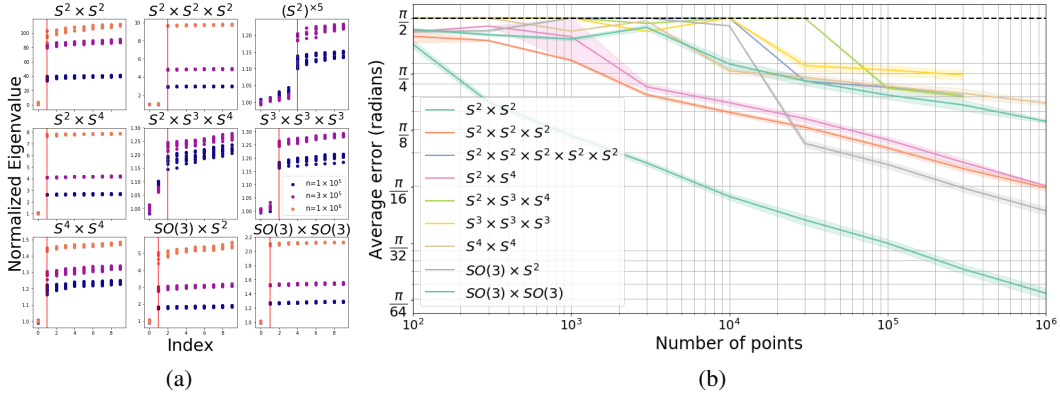

Figure 5: Results on synthetic manifolds. (a) The spectrum of $\overline{\Delta^2}$ for products of spheres and special orthogonal groups with different amounts of data. The spectrum is rescaled so the first eigenvalue equals 1. A clear gap emerges in the spectrum at the eigenvalue equal to the number of submanifolds (red line). (b) The average angle between the subspaces recovered by GEOMANCER and the true tangent spaces of the submanifolds. Past a critical threshold, the error declines with more training data.

onto submanifolds from those that are due to specific properties of the particular manifold. First, many manifolds have eigenfunctions of $\Delta^2$ with zero eigenvalue that are skew-symmetric (Supp. Mat., Sec. C). Moreover, the action of $\Delta^2$ on any function of the form $f_j \mathbf{I}$ will be the same as the action of $\Delta^0$ on $f_j$, meaning eigenvalues of $\Delta^0$ are present in the spectrum of $\Delta^2$ as well. While these will not typically have eigenvalue zero, they may still be small enough to get mixed in with more meaningful results. To avoid both of these spurious eigenfunctions we project each block of $\Delta^2$ onto the space of operators on symmetric zero-trace matrices, to yield a projected block $\overline{\Delta^2_{(ij)}}$ of the projected second-order graph connection Laplacian $\overline{\Delta^2}$. The eigenfunctions of interest can still be expressed as $\sum_j c_j^r \mathbf{\Pi}^{(j)}$ where $\mathbf{\Pi}_i^{(j)}$ is the orthogonal projection onto $T_{\mathbf{x}_i}^{(j)} \mathcal{M}$ and $\sum_j c_j^r \dim(T_{\mathbf{x}_i}^{(j)} \mathcal{M}) = 0$. We can then use standard sparse eigensolvers to find the lowest eigenvalues $\lambda_1, ..., \lambda_R$ and eigenvectors $\phi^1, ..., \phi^R$, which we split into individual vectors $\phi_i^r$ for each point $\mathbf{x}_i$ and project back to full $k \times k$ matrices $\mathbf{\Omega}_i^r$. For details please refer to Sec. C.2.

**Clustering Subspace Dimensions** Once we have computed the smallest eigenvalues $\lambda_1, ..., \lambda_R$ and matrices $\mathbf{\Omega}_1^1, ..., \mathbf{\Omega}_t^R$ from $\overline{\Delta^2}$, we need to merge the results together into a set of orthogonal subspaces $T_{\mathbf{x}_i}^{(1)} \mathcal{M}, ..., T_{\mathbf{x}_i}^{(m)} \mathcal{M}$ at every point. The appropriate number of submanifolds $m$ can be inferred by looking for a gap in the spectrum and stopping before $\lambda_m > \gamma$, similar to how PCA can identify the dimensionality of the best linear projection of a dataset. Due to the degeneracy of the eigenfunctions and the constraint that $\mathrm{tr}(\mathbf{\Omega}_i^r) = 0$, the results will be linear combinations of $\mathbf{\Pi}_i^{(j)}$ that we have to demix. As the projection matrices are orthogonal to one another, they can be expressed in the same orthonormal basis $\mathbf{W}_i$ as $\mathbf{\Pi}_i^{(j)} = \mathbf{W}_i \mathbf{D}_i^{(j)} \mathbf{W}_i^T$, where $\mathbf{D}_i^{(j)}$ are $0/1$-valued diagonal matrices such that $\sum_j \mathbf{D}_i^{(j)} = \mathbf{I}$. This is a simultaneous diagonalization problem, which generalizes eigendecomposition to multiple matrices that share the same eigenvectors. We solve this with the orthogonal FFDIAG algorithm [67], yielding a decomposition $\mathbf{\Omega}_i^r = \mathbf{W}_i \mathbf{\Psi}_i^r \mathbf{W}_i^T$ where $\mathbf{\Psi}_i^r \approx \sum_j c_j^r \mathbf{D}_i^{(j)}$.

The columns of $\mathbf{W}_i$ then need to be clustered, one cluster per disentangled subspace. Let $\psi_{ik}$ be the vector made up of the $k$-th diagonal of $\mathbf{\Psi}_i^r$ for all $r = 1, ..., m-1$. The simultaneous constraints on $c_j^r$ and $\mathbf{D}_i^{(j)}$ push $m-1$-dimensional vectors $\psi_{ik}$ to cluster together in the $m$ corners of a simplex. Thus the vectors can simply be clustered by checking if the cosine similarity between two $\psi_{ik}$ is greater than some threshold. Finally, a basis for every disentangled subspace can be constructed by taking all columns $\mathbf{w}_{ik}$ of $\mathbf{W}_i$ such that $\psi_{ik}$ cluster together. An example is given in Fig. 4.

The complete algorithm for GEOMANCER is summarized in Alg. 1. The basic building blocks are just nearest neighbors, SVD and eigendecomposition. GEOMANCER requires very few hyperparameters – just the dimension $k$, the number of nearest neighbors, and the gap $\gamma$ in the spectrum of $\overline{\Delta^2}$ at

| Object | Latents | | | | LEM d=15 | Pixels | β-VAE d=8 | Chance |
|---|---|---|---|---|---|---|---|---|
| | True | Rotated | Scaled | Linear | | | | |
| Bunny | **0.024** | **0.024** | 0.72±0.68 | 1.42±0.09 | 0.37 | 1.25 | 1.30±0.02 | 1.26±0.23 |
| Dragon | **0.023** | **0.023** | 0.96±0.70 | 1.26±0.37 | 0.32 | 1.26 | 1.15±0.09 | 1.26±0.23 |

Table 1: The average angle between the true disentangled subspaces and the subspaces recovered by GEOMANCER from different embeddings of Stanford 3D objects. The first 5 columns all use information from the true latents, including Laplacian Eigenmaps (LEM) with 15 embedding dimensions, while the next 2 columns only use pixels.

which to stop splitting tangent spaces, which can be chosen by simple heuristics. We demonstrate GEOMANCER's performance in the next section.

## 4 Experiments

To demonstrate the power of GEOMANCER, we investigate its performance on both synthetic manifolds and a dataset of rendered 3D objects. We avoid using existing performance metrics for disentangling, as most are based on the causal or probabilistic interpretation of disentangling [2, 4, 6, 20, 21, 68–70] rather than the symmetry-based one. Furthermore, many disentangling metrics assume that each disentangled factor is one-dimensional [2, 4, 6, 68, 69], which is not the case in our investigation here. Details of dataset generation, training, evaluation and additional results are given in Supp. Mat., Sec. D.

**Synthetic Data**   First, we generated data from a variety of product manifolds by uniformly sampling from either the $n$-dimensional sphere $S^n \subset \mathbb{R}^{n+1}$, represented as vectors with unit length, or the special orthogonal group $SO(n) \subset \mathbb{R}^{n \times n}$, represented as orthonormal matrices with positive determinant. We then concatenated the vectorized data to form a product manifold. As these manifolds are geodesically complete and have a metric structure inherited from their embedding in Euclidean space, we would expect GEOMANCER to discover their global product structure easily.

Due to sampling noise, no eigenvalues were exactly zero, but it can be seen in Fig. 5(a) that the spectrum has a large gap at the number corresponding to the number of submanifolds, which grows with increased data. In Fig. 5(b) we show the average angle between the learned subspaces and the ground truth tangent spaces of each submanifold. For all manifolds, a threshold is crossed beyond which the error begins to decline exponentially with more data. Unsurprisingly, the amount of data required grows for more complex high-dimensional manifolds, but in all cases GEOMANCER is able to discover the true product structure. Even for manifolds like $SO(n)$ that are not simply connected, GEOMANCER still learns the correct local coordinate system. Note that the most complex manifolds we are able to disentangle consist of up to 5 submanifolds, significantly outperforming other recent approaches to symmetry-based disentanglement which have not been applied to anything more complex than a product of two submanifolds [16, 17]. Moreover, our approach is fully unsupervised, while theirs requires conditioning on actions.

**Stanford 3D Objects**   To investigate more realistic data, we applied GEOMANCER to renderings of the Bunny and Dragon object from the Stanford 3D Scanning Repository [71] under different pose and lighting conditions (Fig. 6). We chose to render our own data, as existing datasets for 3D objects are limited to changes in azimuth and a limited range of elevations [6, 34–38]. Instead, we sampled rotations of object pose uniformly from $SO(3)$, while the light location was sampled uniformly from the sphere $S^2$. In Table 1, we show the accuracy of GEOMANCER applied to several different embeddings using the same performance metric as in Fig. 5(b). When applied directly to the true latent state vectors, GEOMANCER performs exceptionally well, even if the state is rotated randomly. When individual dimensions are multiplied by a random scale factor, the performance

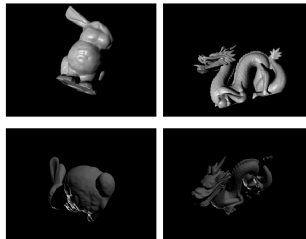

Figure 6: The Stanford Bunny and Dragon with different poses and illumination.

degrades, and if a random linear transformation is applied to the state, performance is no better than chance. This shows that accurate metric information is necessary for GEOMANCER to work. We also applied our method to embeddings learned by Laplacian Eigenmaps using no information other than knowledge of the nearest neighbors in the true latent space. While not as accurate as working from

the true latent state, it still performs far better than chance, showing that metric information alone is *sufficient* for GEOMANCER to work.

Trying to disentangle directly from pixels is more complicated. As the mapping from state to observation is highly nonlinear, GEOMANCER performs no better than chance directly from pixels. However, existing algorithms to disentangle directly from pixels fail as well [19] (See Supp. Mat., Sec. D.3). Even when applying GEOMANCER to the latent vectors learned by the $\beta$-VAE, the results are no better than chance. The poor performance of both GEOMANCER and $\beta$-VAE on the Stanford 3D Objects shows that disentangling full 3D rotations from pixels without side information remains an open problem.

## 5 Discussion

We have shown that GEOMANCER is capable of factorizing manifolds directly from unstructured samples. On images of 3D objects under changes in pose and illumination, we show that correct metric information is critical for GEOMANCER to work. There are several directions for improvement. As GEOMANCER is a nonparametric spectral method, it does not automatically generalize to held-out data, though there are extensions that enable this [72, 73]. While GEOMANCER scales well with the amount of data, the number of nonzero elements in $\overline{\Delta^2}$ grows as $\mathcal{O}(k^4)$ in the dimensionality of the manifold, meaning new approximations are needed to scale to more complex manifolds. The motivating mathematical intuition could also be implemented in a parametric model, enabling end-to-end training.

The missing ingredient for fully unsupervised disentangling is a source of correct metric information. We have shown that existing generative models for disentangling are insufficient for learning an embedding with the right metric structure. We hope that this will be a challenge taken up by the metric learning community. We are hopeful that this charts a new path forward for disentangling research.

### Broader Impact

The present work is primarily theoretical, making its broader impact difficult to ascertain. We consider the algorithm presented here to be a potential core machine learning method, and as such it could have an impact in any area that machine learning can be applied to, but particularly in unsupervised learning, computer vision and robotic manipulation.

### Acknowledgments and Disclosure of Funding

We would like to thank Mélanie Rey, Drew Jaegle, Pedro Ortega, Peter Toth, Olivier Hénaff, Malcolm Reynolds, and Kevin Musgrave for helpful discussions and Shakir Mohamed for support and encouragement.

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
