[Supplementary Material]

# Supplementary Material for "Disentangling by Subspace Diffusion"

## A    Review of Differential Geometry

### A.1    Riemannian manifolds

Consider a $k$-dimensional manifold $\mathcal{M}$. At every point $x \in \mathcal{M}$, the tangent space $T_x\mathcal{M}$ is a $k$-dimensional vector space made up of all velocity vectors $\dot{\gamma}(t)$ where $\gamma : \mathbb{R} \to \mathcal{M}$ is a path such that $\gamma(t) = x$. There are many different ways that a manifold can be embedded in a vector space (for instance, the manifold of natural images can be embedded in the vector space of pixel representations of an image), and quantities on the manifold must be defined in a way that they transform consistently between different embeddings. Let $\mathbf{x} \in \mathbb{R}^n$ be an embedding of the point $x$. Under a differentiable change in embedding $\overline{\mathbf{x}} = f(\mathbf{x})$, tangent vector components $\mathbf{v}$ transform as $\overline{\mathbf{v}} = \mathbf{J}_f \mathbf{v}$, where $\mathbf{J}_f$ is the Jacobian of $f$ at $\mathbf{x}$. The cotangent space $T_x^*\mathcal{M}$ is also a $k$-dimensional vector space, but it consists of all gradients of differentiable functions at $x$ and for finite dimensional manifolds is the dual space to the tangent space. A cotangent vector $\mathbf{w}$ transforms under a change of coordinates as $\mathbf{w} = \mathbf{J}_f^{-1}\mathbf{w}$. When not otherwise specified, we will use "vector" and "tangent vector" interchangeably. Spaces of higher-order tensors can be defined based on how they transform under changes of coordinates. For instance, a linear transform of vectors in $T_x\mathcal{M}$ represented by the matrix $\mathbf{A}$ transforms as $\overline{\mathbf{A}} = \mathbf{J}_f \mathbf{A} \mathbf{J}_f^{-1}$, so linear transforms are rank-(1,1) tensors in $T_x\mathcal{M} \otimes T_x^*\mathcal{M}$.

In a Riemannian manifold, every point $x$ is equipped with a metric $\langle \cdot, \cdot \rangle_x : T_x\mathcal{M} \times T_x\mathcal{M} \to \mathbb{R}$ that defines distances locally. If we choose a basis for the tangent space $T_x\mathcal{M}$, then in that basis the metric can be represented as a positive definite matrix $\mathbf{G}_x \in \mathbb{S}_+^n$ and $\langle \mathbf{v}, \mathbf{w} \rangle_x = \mathbf{v}^T \mathbf{G}_x \mathbf{w}$. This includes the $\ell_2$ metric as a special case when $\mathbf{G}_x = \mathbf{I}$, and has the same form as the Mahalanobis distance from statistics [65], but for tangent vectors instead of distributions. Critically, the metric can change when moving across the manifold. The metric transforms as $\mathbf{G}_{\overline{x}} = \mathbf{J}_f^{-T} \mathbf{G}_x \mathbf{J}_f^{-1}$, so the metric is a rank-(0,2) tensor in $T_x^*\mathcal{M} \otimes T_x^*\mathcal{M} = T_x^{*\otimes 2}\mathcal{M}$. For cotangent vectors, the metric is $\langle \mathbf{v}, \mathbf{w} \rangle_x^* = \mathbf{v}^T \mathbf{G}_x^{-1}\mathbf{w}$, which is a rank-(2,0) tensor.

Once the metric is known in a given coordinate system, the Laplace-Beltrami operator can also be constructed in terms of coordinates:

$$\Delta[f](x) = \frac{1}{\sqrt{\det(\mathbf{G}_x)}} \sum_j \frac{\partial}{\partial x_j} \left( \sqrt{\det(\mathbf{G}_x)} \sum_i g_{ij}^{-1} \frac{\partial f}{\partial x_i} \right) \tag{5}$$

In flat Euclidean space, $\mathbf{G}_x = \mathbf{I}$ and this reduces to the more familiar Laplacian $\Delta[f] = \sum_i \frac{\partial^2 f}{\partial x_i^2}$.

For any two points $x, y \in \mathcal{M}$, the geodesic distance between them is defined as the minimum length of any path between them

$$\mathcal{D}(x,y) = \min_{\substack{\gamma \\ \gamma(0)=x \\ \gamma(1)=y}} \int_0^1 dt \sqrt{\langle \dot{\gamma}(t), \dot{\gamma}(t) \rangle_{\gamma(t)}} \tag{6}$$

A *geodesic* between $x$ and $y$ is a locally shortest path that is parameterised by arc length. In other words it is a path such that there exists a constant $c$ with: $\forall t \in [0,1), \exists \epsilon > 0 | \, \forall t' \in [0,\epsilon] \, \mathcal{D}(\gamma(t), \gamma(t')) = c(t'-t)$. Note that a geodesic is not necessarily a minimum path from start to end. For example, a great circle from the south pole to itself on a sphere is a geodesic even though the distance from the south pole to itself is of course $0$.

534  It's worth noting that the metric is a purely *local* notion of distance, defined only in the tangent space,
535  while the geodestic distance is a *global* distance between two points anywhere on the manifold. Despite
536  the name, the term "metric learning" in machine learning typically refers to learning a single, global
537  notion of distance, or to learning a mapping that *preserves* distances, under the assumption that the
538  correct local distance is already known [66].

## A.2   Parallel transport and affine connections

540  So far we described how to construct a vector space equipped with a metric at every point on the
541  manifold, but have not given any way to relate vectors in one tangent space to those in another. In
542  general there is not a unique mapping from vectors in one tangent space to another, which is precisely
543  why the usual parallelogram model of analogy breaks down when dealing with curved manifolds.
544  Instead, a vector in $T_x\mathcal{M}$ can be identified with a vector in $T_y\mathcal{M}$ in a path-dependent manner through a
545  process called *parallel transport*, where the vector is moved infinitesimally along a path such that it is
546  always *locally* parallel with itself as it moves. To do this, we have to define what it means to be "locally
547  parallel", which requires additional machinery: the *affine connection*.

548  The affine connection at $x$ is a map $\Gamma_x : T_x\mathcal{M} \times T_x\mathcal{M} \to T_x\mathcal{M}$. For two vectors $\mathbf{v}$ and $\mathbf{w} \in T_x\mathcal{M}$,
549  $\Gamma_x(\mathbf{v}, \mathbf{w})$ can be intuitively thought of as the amount the vector $\mathbf{v}$ changes when moving to an
550  infinitesimally nearby tangent space in the direction $\mathbf{w}$. For a Riemannian manifold, there are two
551  natural properties that an affine connection should obey: it should preserve the metric, which means
552  that the inner product between vectors does not change when they are parallel transported, and it should
553  be torsion-free, which intuitively means the vector should not "twist" as it is parallel-transported. Given
554  the appropriate formal definition of these requirements, there is a unique connection that satisfies
555  these properties: the *Levi-Civita* connection. For a given choice of coordinates such that the metric
556  can be represented by $\mathbf{G}_x$ at $x$, and letting the $ij$th element of $\mathbf{G}_x$ be denoted $g_{ij}$ and the $i$th element
557  of $\Gamma_x(\mathbf{v}, \mathbf{w})$ be denoted $\Gamma_x(\mathbf{v}, \mathbf{w})_i$, the Levi-Civita connection at $x$ can be written in terms of the
558  Christoffel symbols

$$\Gamma_x(\mathbf{v}, \mathbf{w})_i = \sum_{jk} \Gamma^i_{jk} v_j w_k$$

$$\Gamma^i_{jk} = \frac{1}{2} \sum_{\ell} g^{-1}_{i\ell} \left( \frac{\partial g_{\ell k}}{\partial x_j} + \frac{\partial g_{\ell j}}{\partial x_k} - \frac{\partial g_{jk}}{\partial x_\ell} \right) \tag{7}$$

559  The Levi-Civita connection defines a *covariant derivative* which takes a vector field $\mathbf{v} : \mathcal{M} \to T_x\mathcal{M}$
560  and a direction $\mathbf{w} \in T_x\mathcal{M}$ and gives the derivative of the field in that direction $\nabla_{\mathbf{w}} \mathbf{v}(x) = \frac{\partial \mathbf{v}}{\partial \mathbf{w}}|_x +$
561  $\Gamma_x(\mathbf{v}(x), \mathbf{w})$. For a manifold embedded in $\mathbb{R}^n$ that also inherits the metric from this space, the covariant
562  derivative is the ordinary derivative in $\mathbb{R}^n$ plus a correction to keep the vector on the manifold, where the
563  affine connection is precisely that correction. In other words, the covariant derivative is the projection
564  of the ordinary derivative onto the manifold. For other Riemannian manifolds, it is better thought of as
565  a correction to force the covariant derivative to transform correctly as a rank-(0,1) tensor. It's worth
566  noting that, as the Levi-Civita connection is a *correction* to make the covariant derivative transform
567  correctly, the connection itself does *not* transform as a tensor. For a change of coordinates $x \to \overline{x}$, the
568  Christoffel symbols transform as:

$$\overline{\Gamma}^i_{jk} = \sum_{mnp} \frac{\partial \overline{x}_i}{\partial x_m} \left[ \Gamma^m_{np} \frac{\partial x_n}{\partial \overline{x}_j} \frac{\partial x_p}{\partial \overline{x}_k} + \frac{\partial^2 x_m}{\partial \overline{x}_j \partial \overline{x}_k} \right] \tag{8}$$

569  where the first term in the sum is the usual change of coordinates for a rank-(1,2) tensor, and the second
570  term is the correction to account for the change in curvature.

571  Informally, two vectors can be thought of as parallel if the covariant derivative in the direction from
572  one to the other is zero. That is, for some infinitesimal $dt$ and vectors $\mathbf{v}, \mathbf{w} \in T_x\mathcal{M}$, the vector
573  $\mathbf{v} + \Gamma_x(\mathbf{v}, \mathbf{w}) dt$ in the tangent space of $x + \mathbf{w} dt$ will be parallel to $\mathbf{v}$. Formally, for a path $\gamma : [0,1] \to \mathcal{M}$,
574  the parallel transport of the starting vector $\mathbf{v}(0) \in T_{\gamma(0)}\mathcal{M}$ is a function $\mathbf{v}(t) \in T_{\gamma(t)}\mathcal{M}$ such that
575  $\nabla_{\dot{\gamma}(t)} \mathbf{v}(t) = \dot{\mathbf{v}}(t) + \Gamma_{\gamma(t)}(\mathbf{v}(t), \dot{\gamma}(t)) = 0$. Parallel transport makes it possible to define a differential
576  equation to solve for the geodesic: if the velocity vector of a path is a parallel transport, that is, if
577  $\nabla_{\dot{\gamma}(t)} \dot{\gamma}(t) = \ddot{\gamma}(t) + \Gamma_{\gamma(t)}(\dot{\gamma}(t), \dot{\gamma}(t)) = 0$, then $\gamma$ is a geodesic. An intuitive way to think of this is that a

578 geodesic is a path that always goes "straight forward" locally – its acceleration is always parallel to the
579 path.

580 Parallel transport can also be defined for higher-order tensors. For a rank-$(p,q)$ tensor $a_{i_1,...,i_p}^{i_1^*,...,i_q^*} \in$
581 $T_x^{\otimes p} \otimes T_x^{*\otimes q}$, the differential equation that defines the parallel transport is given by contracting the
582 Christoffel symbols over all $r$ indices of the tensor:

$$\frac{\partial a_{i_1,...,i_p}^{i_1^*,...,i_q^*}}{\partial t} = \sum_{j_1...j_p,j_1^*...j_q^*,k} \Gamma_{j_1 k}^{i_1}...\Gamma_{j_p k}^{i_p}\Gamma_{i_1^* k}^{j_1^*}...\Gamma_{i_q^* k}^{j_q^*} a_{j_1,...,j_p}^{j_1^*,...,j_q^*}(t)\dot{\gamma}_k(t) \tag{9}$$

## B Definitions and Proofs

584 **Definition 1.** Groups, actions and orbits: A *group* is a set $G = \{g,h,...\}$ equipped with a composition
585 operator $\cdot : G \to G$ such that:

586   1. $G$ is closed under composition: $g \cdot h \in G \ \forall g,h \in G$

587   2. There exists an identity element $e \in G$ such that $g \cdot e = e \cdot g = g \ \forall g \in G$

588   3. The composition operator is associative: $f \cdot (g \cdot h) = (f \cdot g) \cdot h \forall f,g,h \in G$

589   4. For all $g \in G$, there exists an inverse element $g^{-1} \in G$ such that $g \cdot g^{-1} = g^{-1} \cdot g = e$

590 For some other object $z \in Z$, a group *action* is a function $\cdot : G \times Z \to Z$ s.t. $e \cdot z = z$ and $(gh) \cdot z =$
591 $g \cdot (h \cdot z) \forall z \in Z$ and $g,h \in G$. The set $Z$ of all objects under the action of all group elements is referred
592 to as the *orbit* of $z$ under the action of $G$. For instance, the unit sphere is the orbit of a unit vector under
593 all rotations.

594 **Definition 2.** Symmetry-Based Disentangling (Higgins et al., 2018): Let $W$ be the set of world states,
595 $G$ be a group that acts on those world states which factorizes as $G = G_1 \times G_2 \times ... \times G_m$, and $f : W \to Z$
596 be a mapping to a latent representation space $Z$. The representation $Z$ is said to be *disentangled* with
597 respect to the group factorization $G = G_1 \times G_2 \times ... \times G_m$ if:

598   1. There exists an action of $G$ on $Z$.

599   2. The map $f : W \to Z$ is equivariant between the actions of $G$ on $W$ and $Z$, i.e. $g \cdot f(w) =$
600      $f(g \cdot w) \ \forall g \in G, w \in W$, and

601   3. There is a fixed decomposition $Z = Z_1 \times Z_2 \times ... \times Z_m$ such that each $Z_i$ is invariant to the
602      action of $G_j$ for all $j$ except $j = i$.

603 **Theorem.** Main text, Theorem 2: Let $\mathcal{M} = \mathcal{M}_1 \times ... \times \mathcal{M}_m$ be a Riemannian product manifold, and
604 let $T_x^{(1)}\mathcal{M},...,T_x^{(m)}\mathcal{M}$ denote orthogonal subspaces of $T_x\mathcal{M}$ that are tangent to each submanifold.
605 Then the tensor fields $\mathbf{\Pi}^{(i)} : \mathcal{M} \to T_x\mathcal{M} \otimes T_x^*\mathcal{M}$ for $i \in 1,...,m$, where $\mathbf{\Pi}_x^{(i)}$ is the linear projection
606 operator from $T_x\mathcal{M} \to T_x^{(i)}\mathcal{M}$, are in the kernel of the connection Laplacian $\Delta^2$.

607 *Argument.* Given a basis $\mathbf{U}_x^{(i)}$ of the subspace $T_x^{(i)}$, the projection matrix is given by $\mathbf{\Pi}_x^{(i)} = \mathbf{U}_x^{(i)}\mathbf{U}_x^{(i)T}$.
608 As $T_x^{(i)}$ is an invariant subspace under parallel transport, the holonomy of $\mathbf{U}_x^{(i)}$ around any loop
609 has the form $\mathbf{U}_x^{(i)}\mathbf{Q}$ for some orthonormal matrix $\mathbf{Q}$. Therefore the holonomy of $\mathbf{\Pi}_x^{(i)}$ is given by
610 $\mathbf{U}_x^{(i)}\mathbf{Q}\mathbf{Q}^T\mathbf{U}^{(i)T} = \mathbf{U}_x^{(i)}\mathbf{U}^{(i)T} = \mathbf{\Pi}_x^{(i)}$, and $\mathbf{\Pi}_x^{(i)}$ is invariant to parallel transport. As the rate of change
611 for the tensor field $\mathbf{\Pi}^{(i)}$ under diffusion will be 0, the entire tensor field goes to 0 under the action of
612 $\Delta^2$.

613 *Proof.* Each $\mathbf{\Pi}^{(i)}$ is an endomorphism of the tangent bundle. For a general endomorphism $u$ of the
614 tangent bundle, and a general vector field $X$, the covariant derivative satisfies $\nabla(u(X)) = (\nabla u)(X) +$
615 $u(\nabla X)$. Let's replace $u$ with $\mathbf{\Pi}^{(i)}$ in this formula. Since $\mathcal{M}$ is a product of Riemannian manifold, we
616 have that $\nabla(\mathbf{\Pi}^{(i)}(X))$ and $\mathbf{\Pi}^{(i)}(\nabla X)$ are equal. It follows that $\nabla \mathbf{\Pi}^{(i)}$ is always 0, and the Laplacian
617 $\Delta^2 \mathbf{\Pi}^{(i)} = \text{Tr} \nabla^2 \mathbf{\Pi}^{(i)}$ also has to be 0. $\qquad \square$

## C   Spurious Eigenfunctions of $\Delta^2$

### C.1   Analysis

Figure 7: Eigenvalues and eigenfunctions of $\Delta^2$ for the product manifold $S^2 \times S^2 \times S^2$, without restriction to symmetric matrices. The spectrum (left) clearly has 5 nontrivial but small values before the gap. The value of the first 5 nontrivial eigenfunctions at a single point are shown in the remaining figures. The first three are clearly the skew-symmetric volume form, while the remaining two are the expected projection matrices.

Figure 8: Eigenfunctions of $\Delta^2$ for the product manifold $S^3 \times S^3$, without restriction to symmetric matrices. The first nontrivial eigenfunction is the expected projection matrix, while the next eight eigenfunctions are all skew-symmetric – four per manifold.

A complete characterization of the zero eigenfunctions of the second-order connection Laplacian is beyond the scope of this paper. However, we have both empirically and theoretically found several zero eigenfunctions not of the form of projection matrices onto factor manifolds. The spheres $S^2$ and $S^3$ in particular seem to have a zero eigenfunction which maps points on the manifold to a *skew-symmetric* matrix.

In Figs. 7 and 8, we give examples of these eigenfunctions at a random point on $(S^2)^{\times 3}$ and $(S^3)^{\times 2}$. Each submanifold of $S^2$ has a single skew-symmetric eigenfunction, while each submanifold of $S^3$ has four such skew-symmetric eigenfunctions. Looking at the spectrum, the eigenvalues are of similar magnitude to those eigenvalues used by GEOMANCER. Indeed, the individual eigenfunctions separate the submanifolds of interest so cleanly that it is unfortunate that these eigenfunctions do not seem to exist for all manifolds.

For $S^2$ we can construct the skew-symmetric eigenfunction as follows. Let $(\mathbf{v}_1, \mathbf{v}_2)$ be an orthonormal basis for a point on $S^2$, then $\mathbf{v}_1\mathbf{v}_2^T - \mathbf{v}_2\mathbf{v}_1^T$ is a skew-symmetric tensor. This tensor does not depend on the choice of basis, so it is a uniquely defined tensor field on the whole of $S^2$ (This is in fact one way to construct the volume form for $S^2$). As parallel transport preserves orthonormality, this field is left invariant by parallel transport. Any field which is invariant under parallel transport is a zero eigenfunction of the connection Laplacian. For general spheres $S^n$, the volume form will be a rank-$n$ skew-symmetric tensor, and therefore will not in general be an eigenfunction of $\Delta^2$. The interpretation of the 4 skew-symmetric zero eigenfunctions that exist for $S^3$ is still an open question, and we also do not know whether these skew-symmetric eigenfunctions exist for other manifolds.

### C.2   Eliminating Spurious Eigenfunctions

To remove skew-symmetric eigenfunctions, let $\mathbf{\Pi}_{\text{sym}}$ be the linear projection operator from $R^{k \times k}$ to the space of symmetric matrices, which can be represented by a matrix in $\mathbb{R}^{k^2 \times k(k+1)/2}$. Then we can project the blocks of $\Delta^2$ into this smaller space to remove eigenfunctions which are skew-symmetric. Note that this is a projection of full matrices into a lower dimensional space where the matrix is only represented by its upper (or lower) triangular, rather than a projection into the space of full matrices. This has the added benefit of reducing the computational overhead in both space and time by about a factor of 4.

Figure 9: Fraction of points in the training set for which the number and dimensionality of the disentangled subspaces is correctly recovered for synthetic products of spheres and special orthogonal groups. Beyond a critical threshold, the fraction quickly jumps up and plateaus, and on some small manifolds it reaches nearly perfect accuracy.

To avoid eigenfunctions derived from $\Delta^0$, which will always have the form of some scalar function times the identity, we further multiply the blocks by $\mathbf{\Pi}_{\text{tr}} \in \mathbb{R}^{k(k+1)/2 \times k(k+1)/2-1}$, which projects symmetric matrices onto symmetric matrices with zero trace. Putting this all together, we project each block $\Delta^2_{(ij)}$ onto the space of operators on symmetric zero-trace matrices, to yield a projected block $\overline{\Delta^2_{(ij)}} = \mathbf{\Pi}_{\text{tr}}^T \mathbf{\Pi}_{\text{sym}}^T \Delta^2_{(ij)} \mathbf{\Pi}_{\text{sym}} \mathbf{\Pi}_{\text{tr}}$ and projected second-order graph connection Laplacian $\overline{\Delta^2}$.

# D   Experimental Details

For all experiments, we used twice the dimensionality of the manifold for the number of nearest neighbors, and computed the bottom 10 eigenfunctions of $\overline{\Delta^2}$. We chose the threshold $\gamma$ such that the algorithm would terminate at the largest gap in the spectrum. We ran 10 copies of all experiments to validate the robustness of our results. All experiments were run on CPU. The simplest experiments finished within minutes (for instance, $S^2 \times S^2$ with 10,000 data points) while the most complex manifolds required days. The largest experiments, such as $(S^2)^{\times 5}$ with 1,000,000 data points, were terminated after 5 days. On the Stanford 3D Objects data, we implemented some steps in parallel across 100-1000 CPUs, such as computing tangent spaces or connection matrices. This allowed us to complete most steps in GEOMANCER in just a few minutes.

## D.1   Synthetic Manifolds

For the results in Fig. 5(b), we excluded points where the shape of the subspaces was not estimated correctly. In Fig. 9, we count the proportion of points in the training set for which we recovered the correct number and dimensionality of subspaces and find that, past a threshold in the dataset size, the fraction of correct subspace shapes jumps up, and in some cases becomes essentially exact. The fraction of estimated subspaces with the correct shape and the error between those subspaces and the ground truth seem to rise in tandem, which suggests that there is a hard lower limit on the amount of data required for disentangling.

If $\theta_{i,jk} \in (0, \frac{\pi}{2})$ is the largest angle between the ground truth $T^{(j)}_{\mathbf{x}_i}\mathcal{M}$ and the GEOMANCER estimate of $T^{(k)}_{\mathbf{x}_i}\mathcal{M}$ (or $\frac{\pi}{2}$ if the dimensions do not match), then the error in Fig. 5(b) is given as $\frac{1}{t}\sum_{i=1}^{t} \min_{\sigma \in S_m} \frac{1}{m}\sum_{j=1}^{m} \theta_{i,j\sigma_j}$ where the minimum is taken over all permutations of $m$ subspaces.

Figure 10: Additional example renderings of the Stanford Bunny and Stanford Dragon under different pose and illumination conditions.

## D.2 Stanford 3D Objects

A dataset of 100,000 images each of the Stanford Bunny and Stanford Dragon was rendered in MuJoCo [67], originally at 1024x1024 resolution, and downsampled to 64x64 pixels. Images were rendered with a randomly sampled 3D rotation and a randomly sampled illumination source position on a sphere. Latent vectors were represented by a concatenation of unit vectors in $\mathbb{R}^3$ and orthogonal matrices in $\mathbb{R}^{3\times3}$ for a 12-dimensional state vector.

When applying GEOMANCER to data other than the true latent state vectors, we can no longer directly compare against ground truth. Instead, we must align the subspaces around the ground truth data with the subspaces around the training data. Let $\mathbf{z}_1,...\mathbf{z}_t$ be the true latent state vectors and $\mathbf{x}_1,...,\mathbf{x}_t$ be the training data. For each point $\mathbf{x}_i$, we use the basis for the tangent space $\mathbf{U}_{\mathbf{x}_i}$ computed in GEOMANCER, while the tangent space basis $\mathbf{U}_{\mathbf{z}_i}$ for $\mathbf{z}_i$ can be computed in closed form because we know the ground truth is $S^2 \times SO(3)$. Let $i_1,...,i_k$ be the indices of the nearest neighbors of $\mathbf{z}_i$, then we project $\mathbf{z}_{i_1},...,\mathbf{z}_{i_k}$ into the basis $\mathbf{U}_{\mathbf{z}_i}$ and $\mathbf{x}_{i_1},...,\mathbf{x}_{i_k}$ into the basis $\mathbf{U}_{\mathbf{x}_i}$ to form a data matrices $V_{\mathbf{z}_i} = \mathbf{U}_{\mathbf{z}_i}^T (\mathbf{z}_{i_1},...,\mathbf{z}_{i_k})$ and $V_{\mathbf{x}_i} = \mathbf{U}_{\mathbf{x}_i}^T (\mathbf{x}_{i_1},...,\mathbf{x}_{i_k})$. We can then align the two subspaces by computing the orthonormal matrix closest to $\left(V_{\mathbf{z}_i}^T V_{\mathbf{z}_i}\right)^{-1/2} V_{\mathbf{z}_i}^T V_{\mathbf{x}_i} \left(V_{\mathbf{x}_i}^T V_{\mathbf{x}_i}\right)^{-1/2}$ using the same SVD technique used to compute the connection matrices in GEOMANCER. We then compute the angle between ground truth subspaces and subspaces learned by GEOMANCER *after* multiplying by the alignment matrix to give the results in Table. 1.

The different perturbations applied to the data in Table 1 were random orthogonal rotations (Rotated), multiplication by a diagonal matrix with entries sampled from $\exp(\mathcal{N}(0,0.5))$ (Scaled), and multiplication by a random matrix with entries sampled iid from $\mathcal{N}(0,1)$ (Linear). For Laplacian Eigenmaps, two points were considered neighbors if the state vector of one was in the 10 nearest neighbors of the other. Varying numbers of embedding dimensions were used, from 5 to 18, and used as input to GEOMANCER (Fig. 11). Above 13 dimensions, GEOMANCER consistently performs significantly better than chance.

## D.3 Training $\beta$-VAE on Stanford 3D Objects

**Model architecture** We used the standard architecture and optimization parameters introduced in[19] for training the $\beta$-VAE model on the Stanford Bunny and Stanford Dragon datasets. The encoder consisted of four convolutional layers (32x4x4 stride 2, 32x4x4 stride 2, 32x4x4 stride 2, and 32x4x4 stride 2), followed by a 128-d fully connected layer and a 32-d latent representation. The decoder architecture was the reverse of the encoder. We used ReLU activations throughout. The decoder parametrized a Bernoulli distribution. We used Adam optimizer with $1e-4$ learning rate and trained

Figure 11: Results of GEOMANCER trained on embedding from Laplacian Eigenmaps (LEM) with different embedding dimensionalities, using the nearest neighbors from the true latents. While not as accurate as working from the true latents directly, GEOMANCER on LEM embeddings performs significantly better than chance above a certain number of dimensions.

Figure 12: Results of GEOMANCER trained on embedding from $\beta$-VAE with different embedding dimensionalities induced by different values of $\beta$. The $\beta$-VAEs themselves were trained directly from pixels with no knowledge of the true latents. The results are no better than chance.

the models for 1 mln iterations using batch size of 16, which was enough to achieve convergence. The models were trained to optimize the following disentangling objective:

$$\mathcal{L}_{\beta-VAE} = \mathbb{E}_{p(\mathbf{x})}\left[\,\mathbb{E}_{q_\phi(\mathbf{z}|\mathbf{x})}[\log p_\theta(\mathbf{x}|\mathbf{z})] - \beta KL(q_\phi(\mathbf{z}|\mathbf{x}) \,||\, p(\mathbf{z}))\,\right] \qquad (10)$$

where $p(\mathbf{x})$ is the probability of the image data, $q(\mathbf{z}|\mathbf{x})$ is the learned posterior over the latent units given the data, and $p(\mathbf{z})$ is the unit Gaussian prior with a diagonal covariance matrix. For each dataset we trained 130 instances of the $\beta$-VAE with different $\beta$ hyperparameter sampled uniformly from $\beta \in [1,30]$ and ten seeds per $\beta$ setting.

**Model selection** In order to analyze whether any of the trained $\beta$-VAE instances were able to disentangle the two generative subspaces (changes in 3D rotation and lighting), we applied the recently proposed Unsupervised Disentanglement Ranking (UDR) score [61] that measures the quality of disentanglement achieved by trained $\beta$-VAE models by performing pairwise comparisons between the representations learned by models trained using the same hyperparameter setting but with different seeds. This approach requires no access to the ground truth data generative process, and does not make other limiting assumptions that precluded us from applying any other existing disentangling metrics. We used the Spearman version of the UDR score. For each trained $\beta$-VAE model we performed 9 pairwise comparisons with all other models trained with the same $\beta$ value and calculated the corresponding $\text{UDR}_{ij}$ score, where $i$ and $j$ index the two $\beta$-VAE models. Each $\text{UDR}_{ij}$ score is calculated by computing the similarity matrix $R_{ij}$, where each entry is the Spearman correlation between the responses of individual latent units of the two models. The absolute value of the similarity matrix is then taken $|R_{ij}|$ and the final score for each pair of models is calculated according to:

$$\frac{1}{d_a + d_b} \left[ \sum_b \frac{r_a^2 * I_{KL}(b)}{\sum_a R(a,b)} + \sum_a \frac{r_b^2 * I_{KL}(a)}{\sum_b R(a,b)} \right] \tag{11}$$

where $a$ and $b$ index into the latent units of models $i$ and $j$ respectively, $r_a = \max_a R(a,b)$ and $r_b = \max_b R(a,b)$. $I_{KL}$ indicate the "informative" latent units within each model, and $d$ is the number of such latent units. The final score for model $i$ is calculated by taking the median of $\text{UDR}_{ij}$ across all $j$.

**$\beta$-VAE is unable to disentangle Stanford 3D Objects** Fig. 13(a) shows plots UDR scores for the 130 trained $\beta$-VAE models. It is clear that the range of $\beta$ values explored through the hyperparameter search is adequate, since the highest value of $\beta = 30$ resulted in the total collapse of the latent space to the prior (resulting in 0 informative latents for the bunny dataset), and the lowest value of $\beta = 1$ resulted in too many informative latents to represent SO(3)xS2 in a disentangled manner. None of the trained models were able to achieve high UDR scores close to the maximum of 1. The highest UDR scores were achieved by the models with either two or four informative latents, so we visualized whether they were able to learn a disentangled latent representation that factorizes into independent subspaces representing changes in pose and illumination. Fig. 13(b) shows that this is not the case, since manipulations of every latent results in changes in both the position and illumination of the Stanford objects. As a final test we presented the same two models with sets of 100 images of the respective Stanford objects that they were trained on. In each set we fixed the value of one of the object's attributes (pose or illumination), while randomly sampling the other one. A model that is able to disentangle these attributes into independent subspaces should have informative latent dimensions with small variance in their inferred means in the condition where their preferred attribute is fixed. It is clear that no such latents exist for the two $beta$-VAE models, with all informative latents encoding both pose and lighting attributes.

Figure 13: **(a)** Unsupervised Disentanglement Ranking (UDR) scores [61] for 130 $\beta$-VAE models trained with different $\beta$ hyperparameter settings, with ten seeds per setting. UDR scores are plotted against the number of informative latents discovered by the trained model. **(b)** Latent traversals for the $\beta$-VAE models with the highest UDR scores from (a). An initial set of values for the latents is inferred from a seed image, before changing the value of each latent dimension between -2.5 and 2.5 in equal increments and visualizing the corresponding reconstruction. All latents encode changes in both rotation and lighting. **(c)** Inferred means for the informative latents from the $\beta$-VAE models with the highest UDR scores from (a). In each subplot 100 images are presented to the model where the value of one subspace (lighting or rotation) is fixed, while the value of the other subspace is randomly sampled. The plotted inferred means are normalized according to $(\mu_i - \mu)$, where $\mu$ is the mean over 100 inferred means $\mu_i$ for the model latent $i$. If a model learns to disentangle lighting from rotation, then latent dimensions corresponding to each disentangled subspace should show significantly smaller dispersion of inferred means in the condition where the corresponding subspace is fixed. It can be seen that no such latents exist in either of the two $\beta$-VAEs.