[Reviews · NeurIPS 2020]

Review 1

Summary and Contributions: This work proposes a practical method for disentangling, i.e., Geometric Manifold Component Estimator (GEOMANCER). GEOMANCER does not learn a global nonlinear embedding, instead, it learns a set of subspaces to assign to each point, where each subspace is the tangent space of one disentangled submanifold. Thus, GEOMANCER can be used to disentangle manifolds for which there may not be a global axis-aligned coordinate system. Experimental results on both synthetic data and Stanford 3D data are included in this paper. 

Strengths: The intuition here is interesting, i.e., "restrict to Lie groups – groups that are also manifolds, we could use the properties of infinitesimal transformations as a learning signal", and in practice, it seems first perform manifold clustering, then learning the local structure for each sub-manifold, can achieve similar results as GEOMANCER. This work may also bring some new interesting theoretical thinking in this direction. Theorem 2 (section 2), the analysis of the 2nd order Laplacian matrix, is quite informative & as the zero eigenvalue associated with factors of a product manifold, and the eigenfunction for subspace tangent. Experimental results on synthetic data are provided in section 4, and show that GEOMANCER worked well, in particular for orthogonal group SO(n), the proposed algorithms still learns the correct local tangent space. 

Weaknesses: The first step in GEOMANER (see algorithm 1) is to construct NN-graph and then use local PCA to estimate tangent space. If understood correctly, there is no outlier filtering or robust related consideration added here, and raise questions for the robustness of the proposed algo given step 1 is like a foundation for following algos.  Section 3. page 6, "The appropriate number of submanifolds m can be inferred by looking for a gap in the spectrum .... ", estimation of number of submanifolds is interesting and seems one way to evaluate the proposed GEOMANCER, but seems only results in synthetic data is included (e.g., Fig. 5(a)).  Experimental results on Stanford 3D object data (Bunny and Dragon) can be viewed as the key part in this paper to support the claimed contribution. However, it seems GeoManCer only works well when applied to the true latent space vectors, and it fails when applied to pixels. Section 4 also claim the existing algorithms to disentangle directly from pixels fail as well [19], and applying GeoManCer to the latent vectors learned by the β-VAE is not improved either.  Also, it would be better to see the downstream application based on the Disentangling learning results, e.g., classification or cluster (and classification has well defined metric for evaluation), then it can be a better judgement for the proposed contributions. 

Correctness: Yes

Clarity: Yes

Relation to Prior Work: This work did a nice job to cite 60+ related work from disentangling, VAE, CNN, manifold learning, etc. Still, it seems some manifold learning papers are missing here, for example LTSA, Z. Zhang, H. Zha, Principal manifolds and nonlinear dimensionality reduction via tangent space alignment in 2004, this is one of the earliest papers in the machine learning community to directly mention the local tangent space learning. There are also following works after 2004, and some of them are more similar to the proposed GEOMANCER, for example D. Gong, X. Zhao, G. Medioni, Robust Multiple Manifolds Structure Learning, ICML 2012.

Reproducibility: Yes

Additional Feedback: Update --- After discussions with other reviewers & meta-reviewer, we decided not take into account author's response given it is not follow the NeurIPS author response template. I keep my original score for this submission as "6 - marginally above the acceptance threshold".


Review 2

Summary and Contributions: This work presents a new manifold learning method called GeoManCEr that decomposes data into multiple disentangled manifolds. This method is derived similar to Laplacian-based manifold learning methods such as Laplacian eigenmaps and diffusion maps, but here the approximation of the Laplace-Beltrami operator is replaced with an approximation of the second order connection Laplacian. Similar to the mentioned spectral methods, the embedding is performed here via spectral decomposition of the Laplacian. However, in this case the eigenvectors are organized into projection matrices on disentangled tangent spaces, requiring some additional steps to clean spurious eigenvectors (e.g., coming from the "classic" Laplacian operating on functions), cluster together the components of each disentangled manifold, and to organize the resulting coordinates so that each submanifold can select its appropriate tangent spaces and represent data points (and tangent vectors) in them. The method is demonstrated on simple synthetically generated entangled manifolds, where the intrinsic manifold metric is known in advance, as well as a toy example of rendered 3D objects with different rotations and light source positions.

Strengths: This paper provides new insights into the problem of disentangling independent latent factors, viewed here through the lens of factorizing groups of transformations on a data manifold. The authors base their construction on the de Rham decomposition, which itself is based on the holonomy group that considers parallel transport over loops on a manifold. Essentially, the authors seek to extract multiple representations of input data, such as each of them encodes a submanifold with holonomy group independent from all other submanifolds. This provides an important formalism to an important problem that is often ill defined, with mostly heuristic qualitative goals that depend on specific applications rather than studied with rigor. The construction itself here is based on extending the work of Singer and Wu on vector diffusion maps, which enriches more traditional manifold learning by encoding information about tangent spaces and the operation of the connection Laplacian on tangent vector fields. Through careful spectral consideration, the authors identify here the eigenfunctions (or "eigentensors") of the second order connection Laplacian that correspond to projections on tangent spaces of individual disentangled manifolds, and provide a constructive method to extract and cluster them, thus assigning each point multiple representations corresponding to these tangent spaces. An important result established, and verified empirically, is the ability to identify in an unsupervised way the number of disentangled components that should be considered. This is given here by an elegant analogue to the identification of connected components in spectral graph theory. There, the multiplicity of the zero eigenvalue of a graph Laplacian gives the number of connected components, while the corresponding eigenvectors (up to demixing them) identify the association of nodes to components. Similarly, the authors show here that the multiplicity of the zero eigenvalue (or "sufficiently small" eigenvalues, w.r.t. a spectral gap, in practice) provides a reliable indication of the number of submanifolds to consider, while joint decomposition of the corresponding eigentensors together with cosine-similarity clustering could yield reasonable disentangled projections, although it should be noted that this is not shown to work in realistic applications (see weaknesses below).

Weaknesses: The main weakness if the proposed approach here is that it is unclear whether it provides a realistic direction in practical applications. Indeed, while the results shown here for rather simple artificial data seem nice, the application to 3D renderings, which are also rather simple and synthetic, already struggles. The authors do address this point partially by identifying the lack of accurate metric information as a crucial missing ingredient. However, it is not clear how realistic would it be to expect such metric information to be provided, or how sensitive the proposed is to the various artifacts and approximation errors that would clearly be expected in real data. Indeed, data "manifolds" rarely actually correspond to clean manifold models, as they have density variations, dimensionality variations, noise, etc., and much work has been invested in coping with such artifacts. There is also the question of scalability of the proposed method, as it is only demonstrated for very simple examples, but many applications that require disentangling in fact involve much more high dimensional data and complex structure. Realistic and challenging data analysis settings have already been studied extensively within the field of diffusion based manifold learning considered here (e.g., works by R Coifman et al., B Nadler et al., T Berry et al., and A Singer et al., come to mind), and it should be noted in this context that even the VDM approach extended here was developed and successfully applied to challenging tasks in organizing CryoEM data with extremely low SNR. Therefore, the underlying foundations of the proposed approach should provide a sufficient starting point to expect some more promising application. Simply put, if the proposed approach struggles with very simple 3D images with uniformly (densely) sampled variations along two clearly independent manifolds, how would it be realistic to apply in practice?

Correctness: The claims and methodology seem well established, and the authors clearly state some deficiencies and identify one of the main gaps or challenges remaining for practical uses of their proposed approach.

Clarity: The paper is well written. It does require some manifold (non-Euclidean) geometry background to fully understand it at times, but is sufficiently clear given such background.

Relation to Prior Work: The paper provides sufficient background to understand the presented ideas, although it does rely on some prior understanding of nontrivial ideas from differential geometry. The introduction provides reasonable coverage of prior and related work on disentangling. One aspect that could be improved is to provide some preliminaries on diffusion maps and its extensions, which either serve as the foundation for the discussed prior work by Singer and Wu or as related extensions. The authors only mention briefly they extend Laplacian Eigenmaps and Vector Diffusion Maps (VDM), but it would be good to provide further discussion of the works of Coifman and Lafon (ACHA 2016), Nadler et al. (ACHA 2016; NeurIPS 2016), Salhov et al. (ACHA 2012; Machine Learning 2016), Wolf and Averbuch (ACHA 2013), Fan and Zhao (ICML 2019), etc. to present a more complete overview of this well studied field using diffusion in manifold learning, both for scalar functions and for vector fields on Riemannian manifolds. This can also help address the question of how to extract or approximate intrinsic metric information, along the lines of the LEM column shown in Table 1.

Reproducibility: Yes

Additional Feedback: *** Updates following author response *** Unfortunately, the author response cannot be taken into consideration as it does not follow the NeurIPS author response template and does not meet the one-page limit. === Original review === While the focus of the paper seems to be on theoretical aspects, and the motivation for disentangling data manifolds is naturally well understood, it would be good to provide some demonstration of the insights or data organization provided by the application of the proposed approach, rather than just quantify the angle between (tangent) subspaces compared to baselines.


Review 3

Summary and Contributions: This paper proposes an algorithm to construct a space to enable the parallelogram commutation. The commutation may not work in some sub-space, and the algorithm will reorganize the sub-space through the learning component.

Strengths: This work is well-motivated by the fundamental theory of differential geometry, i.e. de Rham decomposition, which motivates the local disentanglement of commutative global manifold. The local disentanglement comes from Theorem 2, which is enabling mechanism in the paper, by finding the objective of Laplacian \triangle^2 to be near zero eigenvalues. Therefore, authors find the disentangled submanifolds, or local coordinates, by the matrix decomposition, i.e. SVD, on data instances, on the second-order connection Laplacian.

Weaknesses: My expertise lies in the disentanglement of the latent spaces with regularization, discriminators, and prior distributions, i.e. \beta-VAE. Therefore, I had to review this paper from the user's perspective. - The proposed method relies on the matrix decomposition on the local PCA estimation, and this happens for all instances. This seems a very arduous computation even with the parallel computing by GPU. Any opinion on this complexity? Simiarily, there are series of matrix multiplications to be reviewed from the computational complexity perspective. - The result is not much supportive considering the result in Table 1 without the true latent information, which limits the practical application of the proposed algorithm.

Correctness: Seems to be correct up to my understanding, but this should be discussed by other reviewers.

Clarity: Yes. The paper itself is clearly written for a researcher with knowledge on linear algebra, abstract vector space, and differential geometry.

Relation to Prior Work: The prior work is not much covered in this line of differential geometry and latent disentanglement. I was not able to find some, either.

Reproducibility: Yes

Additional Feedback: Please focus on the computational complexity question, and I suggest that you create a subsection to discuss the complexity. //// Read the rebuttal. The rebuttal format was not following the Neurips guideline, so the panel decided to ignore the rebuttal.


Review 4

Summary and Contributions: The authors introduce the GeometricManifold Component Estimator (GEOMANCER) which is a really cool name. The provide a partial answer to is it possible to learn how to factorize a Lie group solely from observations of the orbit of an object it acts on? The develop a geometric theory based on holonomy and provide an algorithm based on a discrete Hodge Laplacian which is an approximation of the giving an approximation to the de Rham decomposition from differential geometry. The paper reduces the question of whether unsupervised disentangling is possible to the question of whether unsupervised metric learning is possible, providing a unifying insight into the geometric nature of representation learning.

Strengths: The paper is rigorous, the mathematical ideas are potentially powerful and the exposition outside of the intro and the parallelogram are clear and motivating. The use of de Rham decomposition is nice and appealing. The relations to classic Laplcains is also interesting.

Weaknesses: The paper does not make clear the relation to other works on learning group structure such as The Geometry of Synchronization Problems and Learning Group Actions Tingran Gao, Jacek Brodzki & Sayan Mukherjee Discrete & Computational Geometry (2019) or The Diffusion Geometry of Fibre Bundles: Horizontal Diffusion Maps Tingran Gao. Also the need for the implicit fiber construction over standard diffusion maps was not made clear.

Correctness: Yes,

Clarity: Overall yes. The intro I felt was a bit watered down.

Relation to Prior Work: No as stated above.

Reproducibility: Yes

Additional Feedback:

[Author Response · NeurIPS 2020]

First of all, many thanks to all the reviewers for taking the time to read the paper, and for their thoughtful comments. We'll respond to each point one-by-one. We have color-coded each reviewer's comments for ease of reference, and have highlighted our own responses for clarity.

# Questions

### 1. Summary and contributions: Briefly summarize the paper and its contributions.

This work proposes a practical method for disentangling, i.e., Geometric Manifold Component Estimator (GEOMANCER). GEOMANCER does not learn a global nonlinear embedding, instead, it learns a set of subspaces to assign to each point, where each subspace is the tangent space of one disentangled submanifold. Thus, GEOMANCER can be used to disentangle manifolds for which there may not be a global axis-aligned coordinate system. Experimental results on both synthetic data and Stanford 3D data are included in this paper.

### 2. Strengths: Describe the strengths of the work. Typical criteria include: soundness of the claims (theoretical grounding, empirical evaluation), significance and novelty of the contribution, and relevance to the NeurIPS community.

The intuition here is interesting, i.e., "restrict to Lie groups – groups that are also manifolds, we could use the properties of infinitesimal transformations as a learning signal", and in practice, it seems first perform manifold clustering, then learning the local structure for each sub-manifold, can achieve similar results as GEOMANCER. This work may also bring some new interesting theoretical thinking in this direction. Theorem 2 (section 2), the analysis of the 2nd order Laplacian matrix, is quite informative & as the zero eigenvalue associated with factors of a product manifold, and the eigenfunction for subspace tangent.

Experimental results on synthetic data are provided in section 4, and show that GEOMANCER worked well, in particular for orthogonal group SO(n), the proposed algorithms still learns the correct local tangent space.

### 3. Weaknesses: Explain the limitations of this work along the same axes as above.

The first step in GEOMANER (see algorithm 1) is to construct NN-graph and then use local PCA to estimate tangent space. If understood correctly, there is no outlier

filtering or robust related consideration added here, and raise questions for the robustness of the proposed algo given step 1 is like a foundation for following algos.

We believe that one of the strengths of GEOMANCER is that each step in the algorithm is already a well-understood and well-studied problem (e.g. nearest neighbors, local PCA, block sparse eigendecomposition). This means that any existing algorithms for improving the robustness of any one part of GEOMANCER could easily be dropped in as a replacement for what we are currently using. We want to emphasize that GEOMANCER is not meant to be the end-all algorithm that solve all problems in disentangling - it is only meant as a first demonstration of how to apply the theoretical ideas in the paper (and thank you for the kind words about the interesting new theoretical thinking our paper provides). As the literature on outlier detection and robust PCA is quite large, we felt it was better to start with tried-and-true methods to keep things simple, but more robust extensions would make an excellent follow-up paper, and we thank the reviewer for the suggestion.

Section 3. page 6, "The appropriate number of submanifolds m can be inferred by looking for a gap in the spectrum .... ", estimation of number of submanifolds is interesting and seems one way to evaluate the proposed GEOMANCER, but seems only results in synthetic data is included (e.g., Fig. 5(a)).

We were quite limited for space in the main paper, but we would be happy to add a figure showing the spectrum on the Stanford 3D object data as well in the appendix. We believe that being able to correctly align the different local subspaces is a much harder problem than just identifying the number of submanifolds, so we felt that those were the more important results to show.

Experimental results on Stanford 3D object data (Bunny and Dragon) can be viewed as the key part in this paper to support the claimed contribution. However, it seems GeoManCer only works well when applied to the true latent space vectors, and it fails when applied to pixels. Section 4 also claim the existing algorithms to disentangle directly from pixels fail as well [19], and applying GeoManCer to the latent vectors learned by the β-VAE is not improved either.

The experiments on the Stanford 3D object data was primarily intended to show that metric information is both necessary and sufficient for GEOMANCER to work. This is an important insight that can guide future work on disentangling 3D rotations. We also wanted to demonstrate that existing disentangling methods like beta-VAE fail on this data, showing the need for methods that fuse GEOMANCER with metric learning. As we state in the paper, we only consider GEOMANCER to be halfway to a complete solution to the problem, but even half a solution to such an important problem is an important breakthrough.

Also, it would be better to see the downstream application based on the Disentangling learning results, e.g., classification or cluster (and classification has well defined metric for evaluation), then it can be a better judgement for the proposed contributions.

We agree that it would be nice to demonstrate utility on downstream tasks, though we wanted to keep the paper focused on explaining the core algorithm and saving applications for later work. What we found when looking for good tasks to try GEOMANCER on was that no datasets used in disentangling research actually contain full 3D rotations - they are all restricted to changes in azimuth and small changes in elevation. Therefore we decided it was necessary to generate our own data instead of using existing benchmarks. Already, this dataset has revealed the insufficiency of beta-VAEs for handling full 3D rotations. In the future, we can extend this data to include downstream tasks like abstract visual reasoning, similar to the tasks studied in Steenkiste, Locatello, Schmidhuber and Bachem (2019).

**4. Correctness: Are the claims and method correct? Is the empirical methodology correct?**

Yes

**5. Clarity: Is the paper well written?**

Yes

**6. Relation to prior work: Is it clearly discussed how this work differs from previous contributions?**

This work did a nice job to cite 60+ related work from disentangling, VAE, CNN, manifold learning, etc. Still, it seems some manifold learning papers are missing here, for example LTSA, Z. Zhang, H. Zha, Principal manifolds and nonlinear dimensionality reduction via tangent space alignment in 2004, this is one of the

earliest papers in the machine learning community to directly mention the local tangent space learning. There are also following works after 2004, and some of them are more similar to the proposed GEOMANCER, for example D. Gong, X. Zhao, G. Medioni, Robust Multiple Manifolds Structure Learning, ICML 2012.

Thank you for the kind comments. We did try to be thorough in our literature review, including historical papers that may not be well known in the community. We were unaware of the work on LTSA and RMMSL, but we will happily include those citations. We also want to emphasize that our work is quite different from manifold clustering. Manifold clustering is intended for data that is a union of several manifolds embedded in the same space. GEOMANCER is intended for data that is a direct product of several manifolds - so every data point can be thought of as a concatenation of elements from several manifolds simultaneously, rather than belonging to a single manifold. In that sense, GEOMANCER is really an algorithm for manifold *factorization*, not clustering.

**7. Reproducibility: Are there enough details to reproduce the major results of this work?**

Yes

**9. Please provide an "overall score" for this submission.**

6: Marginally above the acceptance threshold.

**10. Please provide a "confidence score" for your assessment of this submission.**

4: You are confident in your assessment, but not absolutely certain. It is unlikely, but not impossible, that you did not understand some parts of the submission or that you are unfamiliar with some pieces of related work.

**11. Have the authors adequately addressed the broader impact of their work, including potential negative ethical and societal implications of their work?**

Yes

## Questions

**1. Summary and contributions: Briefly summarize the paper and its contributions.**

This work presents a new manifold learning method called GeoManCEr that decomposes data into multiple disentangled manifolds. This method is derived similar to Laplacian-based manifold learning methods such as Laplacian eigenmaps

and diffusion maps, but here the approximation of the Laplace-Beltrami operator is replaced with an approximation of the second order connection Laplacian. Similar to the mentioned spectral methods, the embedding is performed here via spectral decomposition of the Laplacian. However, in this case the eigenvectors are organized into projection matrices on disentangled tangent spaces, requiring some additional steps to clean spurious eigenvectors (e.g., coming from the "classic" Laplacian operating on functions), cluster together the components of each disentangled manifold, and to organize the resulting coordinates so that each submanifold can select its appropriate tangent spaces and represent data points (and tangent vectors) in them. The method is demonstrated on simple synthetically generated entangled manifolds, where the intrinsic manifold metric is known in advance, as well as a toy example of rendered 3D objects with different rotations and light source positions.

**2. Strengths: Describe the strengths of the work. Typical criteria include: soundness of the claims (theoretical grounding, empirical evaluation), significance and novelty of the contribution, and relevance to the NeurIPS community.**

This paper provides new insights into the problem of disentangling independent latent factors, viewed here through the lens of factorizing groups of transformations on a data manifold. The authors base their construction on the de Rham decomposition, which itself is based on the holonomy group that considers parallel transport over loops on a manifold. Essentially, the authors seek to extract multiple representations of input data, such as each of them encodes a submanifold with holonomy group independent from all other submanifolds. This provides an important formalism to an important problem that is often ill defined, with mostly heuristic qualitative goals that depend on specific applications rather than studied with rigor.

The construction itself here is based on extending the work of Singer and Wu on vector diffusion maps, which enriches more traditional manifold learning by encoding information about tangent spaces and the operation of the connection Laplacian on tangent vector fields. Through careful spectral consideration, the authors identify here the eigenfunctions (or "eigentensors") of the second order connection Laplacian that correspond to projections on tangent spaces of individual disentangled manifolds, and provide a constructive method to extract and cluster

them, thus assigning each point multiple representations corresponding to these tangent spaces.

An important result established, and verified empirically, is the ability to identify in an unsupervised way the number of disentangled components that should be considered. This is given here by an elegant analogue to the identification of connected components in spectral graph theory. There, the multiplicity of the zero eigenvalue of a graph Laplacian gives the number of connected components, while the corresponding eigenvectors (up to demixing them) identify the association of nodes to components. Similarly, the authors show here that the multiplicity of the zero eigenvalue (or "sufficiently small" eigenvalues, w.r.t. a spectral gap, in practice) provides a reliable indication of the number of submanifolds to consider, while joint decomposition of the corresponding eigentensors together with cosine-similarity clustering could yield reasonable disentangled projections, although it should be noted that this is not shown to work in realistic applications (see weaknesses below).

### 3. Weaknesses: Explain the limitations of this work along the same axes as above.

The main weakness if the proposed approach here is that it is unclear whether it provides a realistic direction in practical applications. Indeed, while the results shown here for rather simple artificial data seem nice, the application to 3D renderings, which are also rather simple and synthetic, already struggles. The authors do address this point partially by identifying the lack of accurate metric information as a crucial missing ingredient. However, it is not clear how realistic would it be to expect such metric information to be provided, or how sensitive the proposed is to the various artifacts and approximation errors that would clearly be expected in real data. Indeed, data "manifolds" rarely actually correspond to clean manifold models, as they have density variations, dimensionality variations, noise, etc., and much work has been invested in coping with such artifacts. There is also the question of scalability of the proposed method, as it is only demonstrated for very simple examples, but many applications that require disentangling in fact involve much more high dimensional data and complex structure. Realistic and challenging data analysis settings have already been studied extensively within the field of diffusion based manifold learning considered here (e.g., works by R Coifman et al., B Nadler et al., T Berry et al., and A Singer et al., come to mind), and it should

be noted in this context that even the VDM approach extended here was developed and successfully applied to challenging tasks in organizing CryoEM data with extremely low SNR. Therefore, the underlying foundations of the proposed approach should provide a sufficient starting point to expect some more promising application. Simply put, if the proposed approach struggles with very simple 3D images with uniformly (densely) sampled variations along two clearly independent manifolds, how would it be realistic to apply in practice?

First of all, thank you for the very kind words about the importance of both the theoretical framework and the results we have demonstrated empirically. We appreciate that the reviewer has really grasped the importance of our work.

To the point about the practicality of GEOMANCER for real data - we believe that even identifying the source of GEOMANCER's shortcomings on real data provides an important insight that can influence the field in important ways. There is a rich literature on metric learning as well as hand-designed metrics for image data (e.g. SSIM), and we hope that this work can inspire future work on metric learning that treats disentangling as a downstream task. While it is an open question whether this metric information is available in practice, we believe it is a question worth pursuing, and it is hardly a foregone conclusion that this information is not available.

We also want to emphasize that the paper already covers a significant amount of material, and just the theoretical contribution and its demonstration on synthetic data is an important contribution that we believe merits publication. Even the original paper on Neural ODEs was limited to demonstration on synthetic 2D and 3D data, while scaling to practical problems was left to future work like FFJORD. A paper combining the existing work with novel work on metric learning that fully solved the problem of disentangling the Stanford 3D object data would likely be too long and unfocused for a venue like NeurIPS. We felt it was better to stay focused on the theoretical contribution, and discuss limitations as a way of inspiring future work.

**4. Correctness: Are the claims and method correct? Is the empirical methodology correct?**

The claims and methodology seem well established, and the authors clearly state some deficiencies and identify one of the main gaps or challenges remaining for practical uses of their proposed approach.

**5. Clarity: Is the paper well written?**

The paper is well written. It does require some manifold (non-Euclidean) geometry background to fully understand it at times, but is sufficiently clear given such background.

**6. Relation to prior work: Is it clearly discussed how this work differs from previous contributions?**

The paper provides sufficient background to understand the presented ideas, although it does rely on some prior understanding of nontrivial ideas from differential geometry. The introduction provides reasonable coverage of prior and related work on disentangling.

One aspect that could be improved is to provide some preliminaries on diffusion maps and its extensions, which either serve as the foundation for the discussed prior work by Singer and Wu or as related extensions. The authors only mention briefly they extend Laplacian Eigenmaps and Vector Diffusion Maps (VDM), but it would be good to provide further discussion of the works of Coifman and Lafon (ACHA 2016), Nadler et al. (ACHA 2016; NeurIPS 2016), Salhov et al. (ACHA 2012; Machine Learning 2016), Wolf and Averbuch (ACHA 2013), Fan and Zhao (ICML 2019), etc. to present a more complete overview of this well studied field using diffusion in manifold learning, both for scalar functions and for vector fields on Riemannian manifolds. This can also help address the question of how to extract or approximate intrinsic metric information, along the lines of the LEM column shown in Table 1.

Thank you for the suggestions. We do want to emphasize that, while vector diffusion is a well studied topic, ours is to the best of our knowledge the first work on practical applications of subspace/matrix diffusion, and certainly the first to make the connection to disentangling. However, for the sake of completeness, we will expand the discussion of related work to more thoroughly cover the topic of vector diffusion.

**7. Reproducibility: Are there enough details to reproduce the major results of this work?**

Yes

**8. Additional feedback, comments, suggestions for improvement and questions for the authors:**

While the focus of the paper seems to be on theoretical aspects, and the motivation for disentangling data manifolds is naturally well understood, it would be good to provide some demonstration of the insights or data organization provided by the application of the proposed approach, rather than just quantify the angle between (tangent) subspaces compared to baselines.

**9. Please provide an "overall score" for this submission.**

6: Marginally above the acceptance threshold.

**10. Please provide a "confidence score" for your assessment of this submission.**

4: You are confident in your assessment, but not absolutely certain. It is unlikely, but not impossible, that you did not understand some parts of the submission or that you are unfamiliar with some pieces of related work.

**11. Have the authors adequately addressed the broader impact of their work, including potential negative ethical and societal implications of their work?**

Yes

**Reviewer #3**

## Questions

**1. Summary and contributions: Briefly summarize the paper and its contributions.**

This paper proposes an algorithm to construct a space to enable the parallelogram commutation. The commutation may not work in some sub-space, and the algorithm will reorganize the sub-space through the learning component.

**2. Strengths: Describe the strengths of the work. Typical criteria include: soundness of the claims (theoretical grounding, empirical evaluation), significance and novelty of the contribution, and relevance to the NeurIPS community.**

This work is well-motivated by the fundamental theory of differential geometry, i.e. de Rham decomposition, which motivates the local disentanglement of commutative global manifold. The local disentanglement comes from Theorem 2, which is enabling mechanism in the paper, by finding the objective of Laplacian $\triangle^2$ to be near zero eigenvalues. Therefore, authors find the disentangled

submanifolds, or local coordinates, by the matrix decomposition, i.e. SVD, on data instances, on the second-order connection Laplacian.

**3. Weaknesses: Explain the limitations of this work along the same axes as above.**

My expertise lies in the disentanglement of the latent spaces with regularization, discriminators, and prior distributions, i.e. \beta-VAE. Therefore, I had to review this paper from the user's perspective.

- The proposed method relies on the matrix decomposition on the local PCA estimation, and this happens for all instances. This seems a very arduous computation even with the parallel computing by GPU. Any opinion on this complexity? Simiarily, there are series of matrix multiplications to be reviewed from the computational complexity perspective.

We provide a brief discussion on the scaling in the discussion section, but we can expand this in the paper to make it more clear. The matrices we need to decompose for local PCA are actually not very large, so the computational requirements are not too onerous (and because they are small, there is not much advantage to GPU acceleration). We are in fact able to run all of our experiments on CPUs, and the smallest experiments finish in under a minute! Larger experiments, on the scale of the 3D image dataset (O(100k) data points, O(10) latent dimensions) take from hours to days. The main bottleneck to scaling is not the size of the dataset but the latent dimensionality of the manifold - the blocks of the second-order Laplacian are of size k^2-by-k^2 for a k-dimensional manifold, so the overall scaling is O(k^4). This is a limitation of GEOMANCER, but we believe one that can be overcome by smart approximation. Our primary intent in this paper was to explain the theoretical basis for GEOMANCER and present the most straightforward algorithm that could implement this theoretical framework. In the interest of keeping the paper focused, we felt it was best to leave work on better scaling through approximations to future work.

- The result is not much supportive considering the result in Table 1 without the true latent information, which limits the practical application of the proposed algorithm.

This point was also raised by Reviewer #2, and we direct you to our response to their comments.

**4. Correctness: Are the claims and method correct? Is the empirical methodology correct?**

Seems to be correct up to my understanding, but this should be discussed by other reviewers.

**5. Clarity: Is the paper well written?**

Yes. The paper itself is clearly written for a researcher with knowledge on linear algebra, abstract vector space, and differential geometry.

**6. Relation to prior work: Is it clearly discussed how this work differs from previous contributions?**

The prior work is not much covered in this line of differential geometry and latent disentanglement. I was not able to find some, either.

**7. Reproducibility: Are there enough details to reproduce the major results of this work?**

Yes

**8. Additional feedback, comments, suggestions for improvement and questions for the authors:**

Please focus on the computational complexity question, and I suggest that you create a subsection to discuss the complexity.

Thank you for the suggestion. We will expand the brief discussion at the end of the paper into its own section for clarity.

**9. Please provide an "overall score" for this submission.**

6: Marginally above the acceptance threshold.

**10. Please provide a "confidence score" for your assessment of this submission.**

3: You are fairly confident in your assessment. It is possible that you did not understand some parts of the submission or that you are unfamiliar with some pieces of related work. Math/other details were not carefully checked.

**11. Have the authors adequately addressed the broader impact of their work, including potential negative ethical and societal implications of their work?**

Yes

## Questions

**1. Summary and contributions: Briefly summarize the paper and its contributions.**

The authors introduce the GeometricManifold Component Estimator (GEOMANCER) which is a really cool name. The provide a partial answer to is it possible to learn how to factorize a Lie group solely from observations of the orbit of an object it acts on? The develop a geometric theory based on holonomy and provide an algorithm based on a discrete Hodge Laplacian which is an approximation of the giving an approximation to the de Rham decomposition from differential geometry. The paper reduces the question of whether unsupervised disentangling is possible to the question of whether unsupervised metric learning is possible, providing a unifying insight into the geometric nature of representation learning.

**2. Strengths: Describe the strengths of the work. Typical criteria include: soundness of the claims (theoretical grounding, empirical evaluation), significance and novelty of the contribution, and relevance to the NeurIPS community.**

The paper is rigorous, the mathematical ideas are potentially powerful and the exposition outside of the intro and the parallelogram are clear and motivating. The use of de Rham decomposition is nice and appealing. The relations to classic Laplcains is also interesting.

Thank you for the kind comments. We are also glad you like the name.

**3. Weaknesses: Explain the limitations of this work along the same axes as above.**

The paper does not make clear the relation to other works on learning group structure such as The Geometry of Synchronization Problems and Learning Group Actions Tingran Gao, Jacek Brodzki & Sayan Mukherjee Discrete & Computational Geometry (2019) or The Diffusion Geometry of Fibre Bundles: Horizontal Diffusion Maps Tingran Gao. Also the need for the implicit fiber construction over standard diffusion maps was not made clear.

Thank you for the references - we will add them to the paper. Could you clarify what you mean by the need for implicit fiber construction?

**4. Correctness: Are the claims and method correct? Is the empirical methodology correct?**

Yes,

**5. Clarity: Is the paper well written?**

Overall yes. The intro I felt was a bit watered down.

We did want the paper to be accessible to a large audience in the ML community, including people who might be less familiar with differential geometry, but if you have specific suggestions for ways it could be improved without sacrificing accessibility, let us know.

**6. Relation to prior work: Is it clearly discussed how this work differs from previous contributions?**

No as stated above.

**7. Reproducibility: Are there enough details to reproduce the major results of this work?**

Yes

**9. Please provide an "overall score" for this submission.**

7: A good submission; accept.

**10. Please provide a "confidence score" for your assessment of this submission.**

5: You are absolutely certain about your assessment. You are very familiar with the related work.

**11. Have the authors adequately addressed the broader impact of their work, including potential negative ethical and societal implications of their work?**

Yes

[Meta-Review · NeurIPS 2020]

The paper reduces the question of whether unsupervised disentangling is possible to the question of whether unsupervised metric learning is possible, providing a unifying insight into the geometric nature of representation learning. All reviewers think the theory and algorithm developed for decomposing Lie group is novel. The paper is missing citations of previous work related to fibre bundle, and manifold learning, which the authors should remedy in the revised version.